# Androgen-regulated transcription of *ESRP2* drives alternative splicing patterns in prostate cancer

Jennifer Munkley[1†]*, Ling Li[2†], S R Gokul Krishnan[1], Gerald Hysenaj[1], Emma Scott[1], Caroline Dalgliesh[1], Htoo Zarni Oo[3,4], Teresa Mendes Maia[5,6,7,8], Kathleen Cheung[9], Ingrid Ehrmann[1], Karen E Livermore[1], Hanna Zielinska[2], Oliver Thompson[2], Bridget Knight[10], Paul McCullagh[11], John McGrath[12], Malcolm Crundwell[13], Lorna W Harries[2], Mads Daugaard[3,4], Simon Cockell[9], Nuno L Barbosa-Morais[5]*, Sebastian Oltean[2]*, David J Elliott[1]*

[1]Institute of Genetic Medicine, University of Newcastle, Newcastle, United Kingdom; [2]Institute of Biomedical and Clinical Sciences, Medical School, College of Medicine and Health, University of Exeter, Exeter, United Kingdom; [3]Department of Urologic Sciences, University of British Columbia, Vancouver, Canada; [4]Vancouver Prostate Centre, Vancouver, Canada; [5]Instituto de Medicina Molecular João Lobo Antunes, Faculdade de Medicina, Universidade de Lisboa, Lisboa, Portugal; [6]VIB Center for Medical Biotechnology, VIB, Ghent, Belgium; [7]VIB Proteomics Core, VIB, Ghent, Belgium; [8]Department for Biomolecular Medicine, Ghent University, Ghent, Belgium; [9]Bioinformatics Support Unit, Faculty of Medical Sciences, Newcastle University, Newcastle, United Kingdom; [10]NIHR Exeter Clinical Research Facility, Royal Devon and Exeter NHS Foundation Trust, Exeter, United Kingdom; [11]Department of Pathology, Royal Devon and Exeter NHS Foundation Trust, Exeter, United Kingdom; [12]Exeter Surgical Health Services Research Unit, Royal Devon and Exeter NHS Foundation Trust, Exeter, United Kingdom; [13]Department of Urology, Royal Devon and Exeter NHS Foundation Trust, Exeter, United Kingdom

**\*For correspondence:**
Jennifer.munkley@ncl.ac.uk (JM);
nmorais@medicina.ulisboa.pt
(NLB-M);
s.oltean@exeter.ac.uk (SO);
David.Elliott@ncl.ac.uk (DJE)

[†]These authors contributed equally to this work

**Competing interests:** The authors declare that no competing interests exist.

**Abstract** Prostate is the most frequent cancer in men. Prostate cancer progression is driven by androgen steroid hormones, and delayed by androgen deprivation therapy (ADT). Androgens control transcription by stimulating androgen receptor (AR) activity, yet also control pre-mRNA splicing through less clear mechanisms. Here we find androgens regulate splicing through AR-mediated transcriptional control of the epithelial-specific splicing regulator *ESRP2*. Both *ESRP2* and its close paralog *ESRP1* are highly expressed in primary prostate cancer. Androgen stimulation induces splicing switches in many endogenous ESRP2-controlled mRNA isoforms, including splicing switches correlating with disease progression. *ESRP2* expression in clinical prostate cancer is repressed by ADT, which may thus inadvertently dampen epithelial splice programmes. Supporting this, treatment with the AR antagonist bicalutamide (Casodex) induced mesenchymal splicing patterns of genes including *FLNB* and *CTNND1*. Our data reveals a new mechanism of splicing control in prostate cancer with important implications for disease progression.
DOI: https://doi.org/10.7554/eLife.47678.001

## Introduction

Prostate is the most common male sex-specific cancer (*Center et al., 2012*). Prostate cancer progression is controlled by androgen steroid hormones including testosterone and its active

**eLife digest** Cancers often begin as cells that grow in connected sheets or clumps known as epithelial cells. To spread, the cancer cells need to change into cells that can break away from the group and move through the tissues. In prostate cancer, this process can happen years after successful treatment, but researchers are not sure why.

Prostate cancer grows in response to testosterone. This hormone circulates around the body, and when it goes into a cell it helps select which genes are switched on or off. Testosterone-blocking drugs can help slow prostate cancer growth by changing this switching on and off of genes. But, over time, some cancers become resistant to the effects of these drugs and start to spread. This may be down to complexities in how testosterone controls gene activity.

To produce a protein, a human cell first makes a copy of the corresponding gene. This copy is then modified, cutting and pasting different parts of the sequence (a process called 'splicing') before the protein is produced. The patterns of splicing a cell exhibits depend on splicing regulator proteins.

Testosterone can change splicing patterns in prostate cancer cells, but researchers did not know how. To find out, Munkley et al. examined a set of genes that turn off in response to testosterone-blocking drugs in people with prostate cancer. This revealed that testosterone controls a master splicing regulator called ESRP2, which is normally present in epithelial cells. In prostate cancer cells in mice, extra ESRP2 slowed tumour growth. But, although ESRP2 levels are high in human prostate cancer cells to begin with, they drop in response to testosterone-blocking drugs. In the laboratory grown cells, the result was a switch away from 'epithelial-like' gene splicing patterns. Some of the new splicing patterns correlated with better patient prognosis, but other splicing patterns might help cancer cells to spread around the body.

These results raise the possibility that blocking testosterone may impair prostate cancer growth, but also inadvertently prepare cancer cells to break away from tumours. A more complete understanding of how testosterone controls splicing could help explain why some tumours initially shrink when testosterone is blocked, but then later spread. Identifying the genes controlled by ESRP2 may reveal new drug targets to improve prostate cancer treatment.
DOI: https://doi.org/10.7554/eLife.47678.002

metabolite 5-α dihydroxytestosterone. Androgens stimulate androgen receptor (AR) signalling in prostate cancer cells to control transcription, including of genes that regulate the cell cycle, central metabolism and biosynthesis, as well as housekeeping functions (*Livermore et al., 2016*). The roles of both androgens and the AR in transcription have been intensively investigated. However, androgens and the AR also regulate alternative pre-mRNA splicing through still largely unknown mechanisms (*Munkley et al., 2018*; *Rajan et al., 2011*). This represents a very important knowledge gap: alternative splicing patterns in cancer cells can generate protein isoforms with different biological functions (*Oltean and Bates, 2014*), and is a key process in the generation of biological heterogeneity in prostate cancer (*Rajan et al., 2009*).

Androgens are also closely linked to prostate cancer treatment, with androgen deprivation therapy (ADT) being the principal pharmacological strategy for locally advanced and metastatic disease. ADT utilises drugs to inhibit gonadal and extra-gonadal androgen biosynthesis and competitive AR antagonists to block androgen binding and abrogate AR function (*Livermore et al., 2016*). ADT delays disease progression, but after 2–3 years tumours often grow again, developing castration resistance with a median survival time of 16 months (*Karantanos et al., 2013*). The central role of androgens and the AR in prostate cancer, and the poor clinical outlook of castration-resistance prostate cancer (CRPCa), have made it crucially important to identify androgen-regulated target genes and mechanisms of function –particularly those that relate to metastasis. The process of epithelial-mesenchymal transition (EMT) plays a pivotal role in prostate cancer metastasis (*Gravdal et al., 2007*; *Matuszak and Kyprianou, 2011*; *Min et al., 2010*; *Saini et al., 2011*; *Xie et al., 2010*). While the mechanisms driving EMT in prostate cancer are poorly understood, ADT has recently been shown to directly induce EMT in both mouse and human prostate tissue (*Sun et al., 2012*;

*Zhifang et al., 2015*). Importantly, changes in alternative splicing patterns can have dramatic effects on EMT and on metastatic disease progression (*Pradella et al., 2017*).

While the mechanisms through which androgens regulate splicing control are not well understood, splicing itself takes place in the spliceosome, which is a multi-component structure containing a core of essential proteins and small nuclear RNAs (*Papasaikas and Valcárcel, 2016*). Splicing inclusion of alternative exons is often controlled by splicing regulator proteins that bind either to regulated exons or within their adjacent flanking intron sequences (*Gabut et al., 2008*). The estrogen and progesterone steroid nuclear hormone receptors control splicing via recruitment of alternative splicing regulators (including the RNA helicases Ddx5 and Ddx17) (*Auboeuf et al., 2007*; *Auboeuf et al., 2004*; *Auboeuf et al., 2002*), and by changing RNA polymerase II extension rates and chromatin structure to affect splice site selection (*Kornblihtt et al., 2009*; *Naftelberg et al., 2015*). Steroid hormones can also drive selection of alternative promoters to include different upstream exons in mRNAs (*Dutertre et al., 2010*; *Munkley et al., 2018*). However, to what extent the above mechanisms may contribute to androgen-mediated splicing is largely unknown.

We reasoned that a potential model to unify the role of androgens and the AR in transcription and splicing control could be via transcriptional regulation of genes that encode splicing regulatory proteins.To address this we analysed a recently described set of genes that reciprocally change expression in response to androgen stimulation in culture and ADT in patients (*Munkley et al., 2016*). Here we identify AR-mediated transcriptional control of the key splicing regulator protein Epithelial Splicing Regulator Protein 2 (ESRP2). Importantly, many ESRP2-regulated exons switch splicing in response to androgen stimulation. ESRP2 and its close relative ESRP1 (60% identical to ESRP2 protein) are important regulators of epithelial alternative splicing patterns (*Bebee et al., 2015*; *Kalluri and Weinberg, 2009*; *Oltean and Bates, 2014*; *Valastyan and Weinberg, 2011*; *Warzecha et al., 2010*; *Warzecha et al., 2009a*; *Warzecha et al., 2009b*), reduced expression of which can drive critical aspects of EMT (*Hayakawa et al., 2017*; *Pradella et al., 2017*; *Warzecha et al., 2010*). Our data identify an AR-ESRP2 axis controlling splicing patterns in prostate cancer cells, and further suggest that reduced ESRP2 levels in response to ADT may inadvertently help prime prostate cancer cells to facilitate longer term disease progression.

## Results

### *ESRP2* is a direct target for AR regulation in prostate cancer cells

To first gain insight into how androgens may mediate patterns of splicing control, we analysed a recently generated dataset of genes that exhibit reciprocal expression patterns on acute androgen stimulation in vitro versus clinical ADT (*Munkley et al., 2016*). While a number of genes encoding splicing factors changed expression in response to acute androgen stimulation in vitro, *ESRP2* also showed a reciprocal expression switch between acute androgen stimulation in culture and ADT in patients (*Munkley et al., 2016*). *ESRP2* expression decreased following ADT in 7/7 prostate cancer patients (*Rajan et al., 2014*) (*Figure 1A*). Furthermore, RNAseq data prepared from different stages of LTL331 patient-derived xenografts (*Akamatsu et al., 2015*) showed reduced *ESRP2* mRNA levels following castration and relapse neuroendocrine prostate cancer (NEPC, *Figure 1B*). We similarly analysed expression of *ESRP1*. *ESRP1* is a close paralog of *ESRP2*, but was not identified in our initial screen to identify androgen-regulated genes (*Munkley et al., 2016*). *ESRP1* expression levels also reduced following ADT (*Figure 1A*). However, *ESRP1* showed less change in gene expression compared to *ESRP2* in patient-derived xenografts following castration or relapse NEPC (*Figure 1C*) (*Akamatsu et al., 2015*).

Further analyses supported androgen-mediated control of *ESRP2* but not *ESRP1* in prostate cancer cell lines. Western blots detected high endogenous levels of both ESRP1 and ESRP2 proteins within the AR positive LNCaP and CWR22 RV1 prostate cancer cell lines, as compared to the AR negative PC3 and DU145 prostate cancer cell lines (*Figure 1D and E*). However, qPCR analysis showed that while androgens activated *ESRP2* gene expression in response to in AR-positive LNCaP cells, this was not observed for *ESRP1* gene expression (*Figure 1F*). Androgen mediated-control of *ESRP2* expression was also detected in two additional AR-expressing prostate cell lines (VCaP and RWPE-1, *Figure 1G*). ESRP2 protein expression was detected 48 hr after androgen exposure, with ESRP1 protein levels not changing over this same time-period (*Figure 1H*). The specificity of the

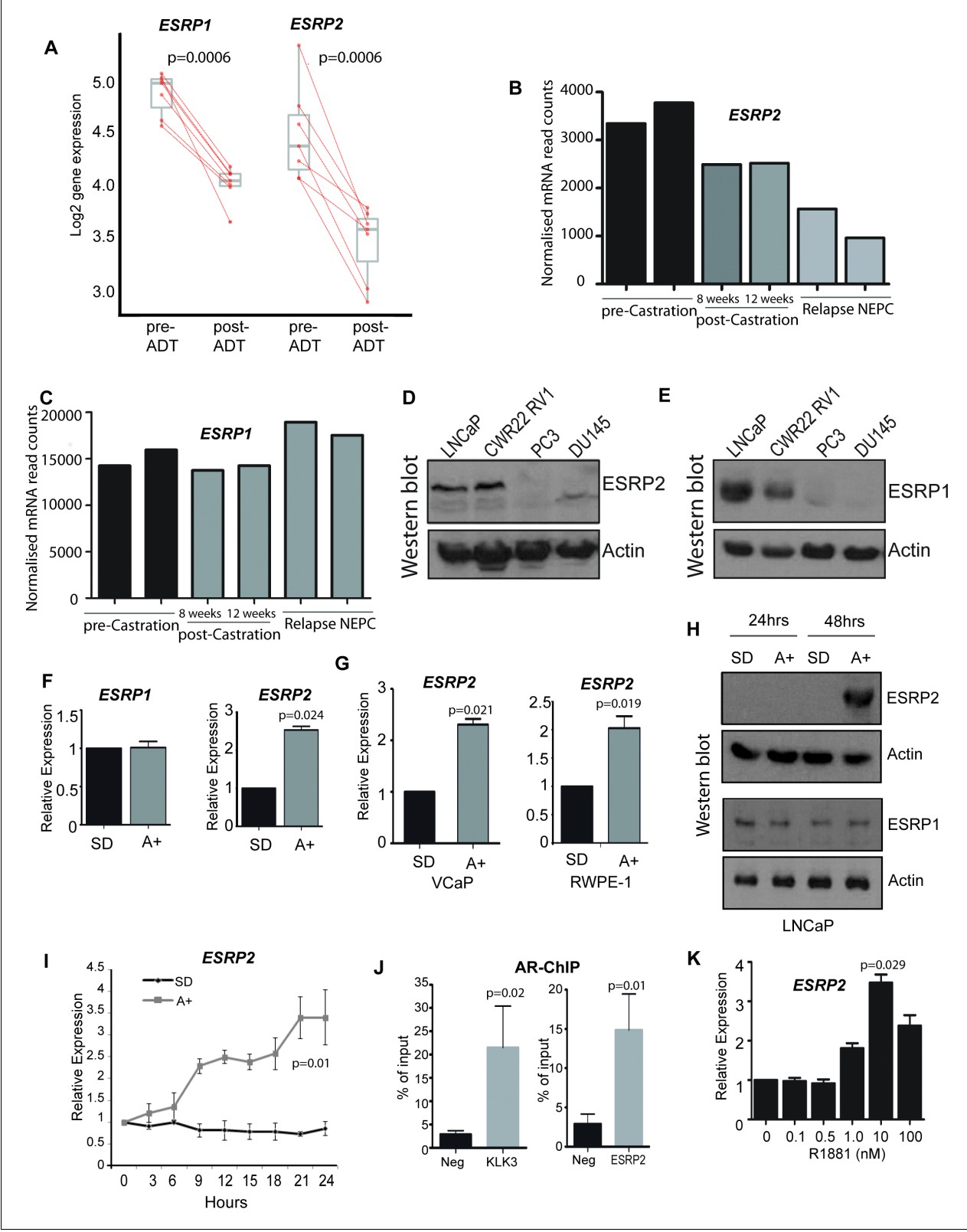

**Figure 1.** *ESRP2* is a direct target for AR regulation in prostate cancer cells. (**A**) Analysis of RNAseq data from human prostate cancer pre- and post-androgen deprivation therapy (ADT) (*Chen et al., 2018*; *Rajan et al., 2014*) shows that there is a significant downregulation of ESRP1 and *ESRP2* mRNA following ADT in all seven patients tested (p=6e-04, Mann Whitney U test). (**B–C**) RNAseq data from LTL331 patient-derived xenografts grown in mice (*Akamatsu et al., 2015*) show a greater reduction in (**B**) *ESRP2* mRNA levels following castration compared to (**C**) ESRP1 mRNA levels. (**D**)

*Figure 1 continued on next page*

*Figure 1 continued*

Western blot analysis of ESRP2 levels in a range of prostate cancer cell lines (actin was used as a loading control). (E) Western blot analysis of ESRP1 levels in prostate cancer cell lines. (F) Real-time PCR analysis of *ESRP2* and *ESRP1* mRNAs in LNCaP cells grown in steroid deplete (SD) or androgen (A +) treated conditions for 24 hr (statistical significance calculated by t test). (G) Real-time PCR analysis of ESRP2 mRNA in RWPE-1 cells grown in steroid deplete (SD) or androgen (A+) treated conditions for 24 hr. (H) Western blots analysis of ESRP1 and 2 protein in LNCaP cells treated with 10nm R1881 (androgens) for 24 and 48 hr. (I) Quantitative analysis (real-time PCR) of *ESRP2* mRNA expression over a 24 hr time course following androgen exposure. (J) Real-time PCR analysis of AR-ChIP performed in LNCaP cells treated with 10nM R1881 for 24 hr revealed AR binding proximal to the *ESRP2* gene. (K) Induction of *ESRP2* is evident in LNCaP cells treated with R1881 concentrations between 1 to 100 nM (p value of 0.029 is for the comparison between 0 nm and 10nm R1881). Statistical significances were calculated by t tests, apart from (A) which used a Mann Whitney U test, and H which used Two-way ANOVA. Real time PCR analyses used at least three independent biological replicates (RNA prepared from separate samples), apart from the AR ChIP (panel I) for which each value shown is a mean of 3 technical replicates.

DOI: https://doi.org/10.7554/eLife.47678.003

The following figure supplement is available for figure 1:

**Figure supplement 1.** Confirmation of the specificity of antibodies against ESRP1 and ESRP2.

DOI: https://doi.org/10.7554/eLife.47678.004

ESRP1 and ESRP2 antibodies used in these experiments was confirmed by detection of over-expressed protein and detection of siRNA mediated protein depletion by western blot (*Figure 1—figure supplement 1*).

Further experimental analyses also supported *ESRP2* as an early and so likely direct target for transcriptional control by the AR: (i) *ESRP2* gene expression in LNCaP cells was rapidly induced in response to 10 nM of the synthetic androgen analogue R1881 (methytrienolone) (*Figure 1I*). (ii) Chromatin immunoprecipitation (ChIP) from LNCaP cells confirmed direct AR binding to a site within 20 Kb of the *ESRP2* gene promoter that had been previously predicted from a genome-wide study (at position chr16: 68210834–68211293 on human genome assembly hg38) (*Massie et al., 2011*) (*Figure 1J*). The AR ChIP signal adjacent to *ESRP2* was similar to that detected in parallel for *KLK3* (encoding prostate specific antigen, or PSA), which is a known AR-regulated gene. (iii) Consistent with *ESRP2* regulation at physiological androgen concentrations, *ESRP2* transcription in LNCaP cells was induced over a wide range of R1881 concentrations, ranging from 1 nM to 100 nM (*Figure 1K*). Each of these above data are consistent with AR-mediated regulation of *ESRP2* expression levels within prostate cancer cell lines as well as tissue.

## *ESRP2* and its paralog *ESRP1* are highly expressed in primary prostate tumours and inhibit tumour growth in vivo

We next monitored *ESRP1* and *ESRP2* expression profiles from prostate cancer patients. Meta-analysis of 719 clinical prostate cancer tumours from 11 previously published studies detected significant up-regulation of both *ESRP1* and *ESRP2* in 9/11 datasets (*Figure 2—source data 1*) (*Arredouani et al., 2009*; *Cancer Genome Atlas Research Network, 2015*; *Fraser et al., 2017*; *Grasso et al., 2012*; *Lapointe et al., 2004*; *Liu et al., 2006*; *Luo et al., 2002*; *Taylor et al., 2010*; *Tomlins et al., 2007*; *Vanaja et al., 2003*; *Varambally et al., 2005*; *Wallace et al., 2008*). We experimentally validated this meta-analysis using two independent panels of clinical samples. Real-time PCR showed significant up-regulation of both *ESRP1* and *ESRP2* mRNA in (1) prostate carcinoma relative to benign prostate hyperplasia (BPH) (*Figure 2A*); and (2) in nine prostate tumour samples relative to matched normal tissue from the same patient (*Figure 2B*). A recent study by *Walker et al. (2017)* identified a molecular subgroup of prostate cancers with metastatic potential at presentation. Within this dataset *ESRP1* was 2.76 fold up-regulated in the 'metastatic-subgroup' compared to the 'non-metastatic subgroup'. Using RNA from a subset of samples from the Walker et al. study, we confirmed significant (p<0.05) upregulation of the *ESRP1* gene in primary prostate cancer patients presenting with a metastatic biology (*Figure 2C*). *ESRP2* gene expression did not significantly increase in the 20 samples studied. We also used these same samples to assess if the observed up-regulation of *ESRP1* and *ESRP2* could result from prostate tumours consisting of a more pure population of epithelial-derived cells compared to matched tissue. Arguing against this possibility, levels of E-Cadherin were not significantly increased between BPH compared to prostate carcinoma, or between matched tumour and normal prostate tissue from patients (*Figure 2—figure supplement 1*).

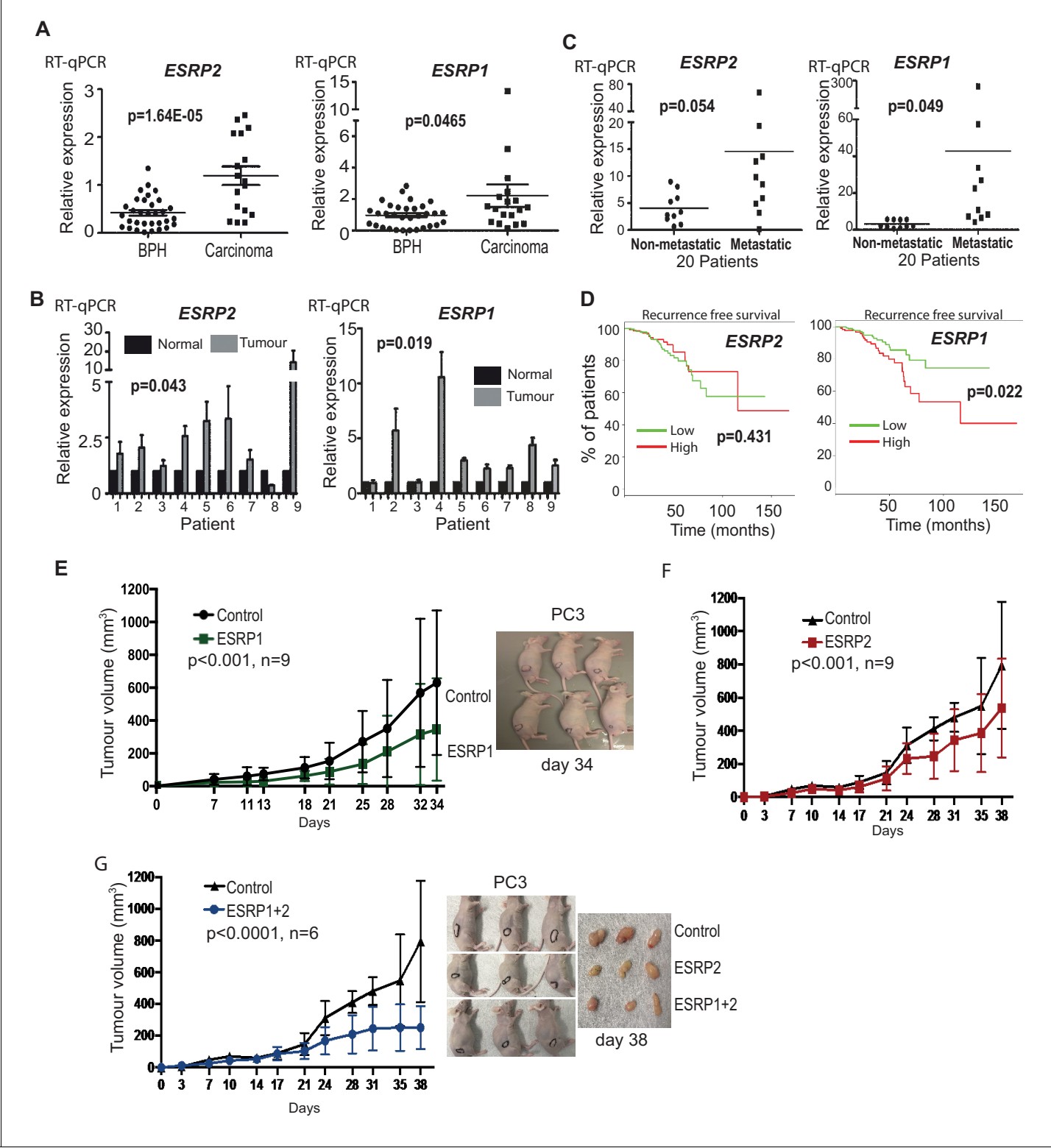

**Figure 2.** ESRP2 and its paralog ESRP1 are highly expressed in primary prostate tumours. (**A**) Real-time PCR analysis of *ESRP1* and *ESRP2* mRNA from patients with benign prostate hyperplasia (BPH) and 17 malignant samples from transurothelial resection of the prostate (TURP) samples. (**B**) Real-time PCR analysis of *ESRP1* and *ESRP2* mRNA from normal and matched prostate cancer tissue from nine patients obtained from radical prostatectomy. (**C**) Analysis of *ESRP1* and *ESRP2* mRNA levels in samples from the *Walker et al. (2017)* cohort. Statistical analysis in parts (**A**)-(**C**) were performed using t tests. (**D**) Interrogation of the TCGA PRAD (PRostate ADenocarcinoma) cohort using KM-express (*Chen et al., 2018*). *ESRP1* expression levels linked to

*Figure 2 continued on next page*

*Figure 2 continued*

a reduced time to PSA biochemical recurrence (bifurcate gene expression at average, log-rank test p=0.022). Over-expression of (**E**) ESRP1, (**F**) ESRP2, or (**G**) both ESRP1 and ESRP2 in PC3 cells significantly slowed the growth of prostate cancer xenografts in vivo. Data were analysed by Two-way ANOVA, and the p value is for the overall difference between two groups.

DOI: https://doi.org/10.7554/eLife.47678.005

The following source data and figure supplements are available for figure 2:

**Source data 1.** Meta-analysis of 719 clinical prostate cancer tumours from 11 previously published studies detected significant up-regulation of both *ESRP1* and *ESRP2* in 9/11 datasets.

DOI: https://doi.org/10.7554/eLife.47678.008

**Figure supplement 1.** E-Cadherin levels are not significantly increased within primary prostate tumours.

DOI: https://doi.org/10.7554/eLife.47678.006

**Figure supplement 2.** Ectopic expression of ESRP1 and ESRP2 protein expression in AR negative cells.

DOI: https://doi.org/10.7554/eLife.47678.007

Each of the above data showed that *ESRP1* and *ESRP2* expression levels are relatively high in primary prostate cancer compared to normal prostate tissue. High *ESRP2* expression was not prognostic of disease progression in the TCGA (PRostate ADenocarcinoma) PRAD cohort analysed using KM-express (*Chen et al., 2018*), but high expression of *ESRP1* associated with a significantly reduced time to first biochemical recurrence (p=0.022) (*Figure 2D*). We tested our antibodies against ESRP1 and ESRP2 proteins on prostate cancer FFPE tissue and cell blocks, but they did not pass our stringent quality control tests (*Figure 1—figure supplement 1C*). While this manuscript was in preparation, another group used an alternative ESRP1 antibody to show up-regulation of ESRP1 in 12,000 prostate cancer tissue microarray tumours (*Gerhauser et al., 2018*).

We next investigated the effects of ESRP1/2 expression on the biology of prostate cancer cells in vivo. Because of their low normal endogenous expression profiles (*Figure 1C and D*), we selected PC3 and DU145 cells to study the effects on prostate cancer cells of ESRP1/ESRP2 protein up-regulation. Ectopic expression of ESRP1 and ESRP2 protein expression in AR negative PC3 and DU145 cell line models reduced prostate cancer cell growth in vitro (*Figure 2—figure supplement 2*). Over-expression of both ESRP1 and ESRP2 (either alone or together) in PC3 cells also significantly slowed growth of prostate cancer xenografts in vivo (*Figure 2E–G*). Taken together, the above data show that ectopic expression of ESRP1 and ESRP2 proteins slow the growth of PC3 and DU145 prostate cancer cell lines and are strongly suggestive that high levels of ESRP2 protein inhibit growth of prostate cancer cells.

## Identification of endogenous ESRP1/ESRP2-regulated targets in prostate cancer cells

To enable us to test whether androgens may control splicing indirectly via transcriptional regulation of *ESRP2*, we next set out to identify a panel of endogenous ESRP2-responsive exons within prostate cancer cells. We first used siRNAs to jointly deplete both ESRP1 and ESRP2 proteins from LNCaP cells (since ESRP1 and ESRP2 can regulate overlapping targets); and in parallel treated LNCaP cells with a control siRNA. We then used RNAseq to monitor the effects of these treatments on the LNCaP transcriptome. Bioinformatic analysis (*Trincado et al., 2018*) of these RNAseq data (GSE129540) predicted 446 ESRP1/ESRP2 regulated alternative splicing events across 319 genes (ΔPSI > 10%, p<0.05) (*Figure 3—source data 1*). We experimentally validated splicing switches for 44 predicted ESRP1/ESRP2-controlled exons by RT-PCR analysis, after LNCaP cells were treated with either of two independent sets of siRNAs directed against ESRP1 and ESRP2 or control siRNAs (*Figure 3* and *Figure 3—source data 2*). We detected similar splicing switches for 35/44 of these skipped exons after jointly depleting ESRP1 and 2 from the AR-positive CWR22 RV1 prostate cancer cell line. 28/44 of these splicing switches were also observed after jointly depleting ESRP1 and ESRP2 from the AR positive PNT2 cells that model the normal prostate epithelium (*Figure 3* and *Figure 3—source data 2*).

Given this set of endogenous target exons, we carried out further analyses to next identify target exons that respond to increasing levels of either ESRP2 or ESRP1 expression in PC3 cells (which normally express low levels of endogenous ESRP1/ESRP2) (*Figure 1D and E*). Ectopic expression of either ESRP1 or ESRP2 in PC3 cells induced splicing switches for 31/42 exons analysed. Importantly,

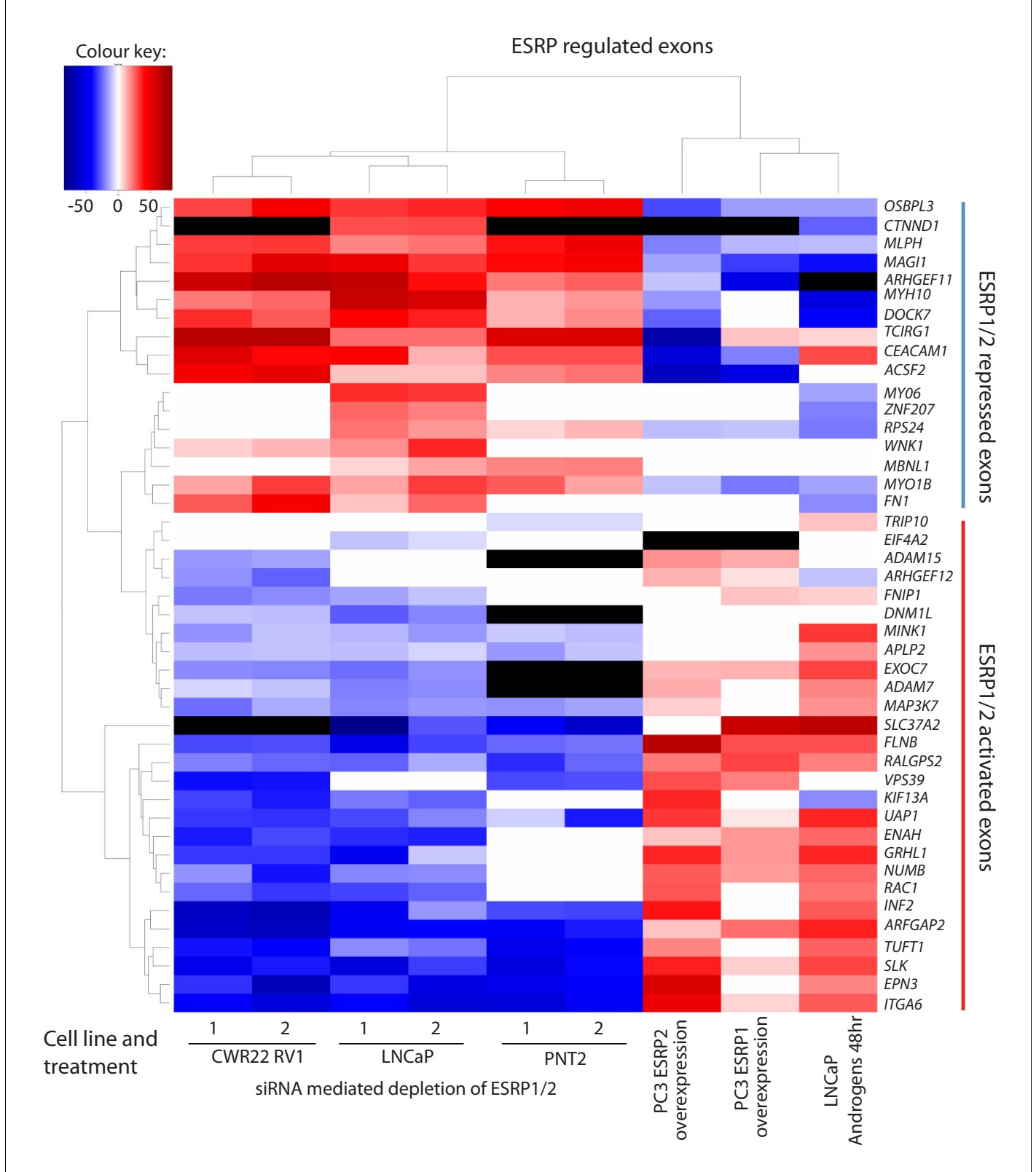

**Figure 3.** Identification of endogenous ESRP1/ESRP2 regulated target exons in prostate cancer. Heat map showing mean PSI levels for a panel of ESRP-regulated exons in prostate cancer cell lines (CWR22RV1, PNT2, LNCaP and PC3). Mean PSIs were calculated for ESRP-regulated isoforms between cells treated with siRNAs specific to ESRP1 and ESRP2, or control siRNAs (CWR22RV1, PNT2, LNCaP), between PC3 cells with and without ectopic expression of ESRP1 or ESRP2, and between LNCaP cells grown in steroid deplete versus androgen stimulated conditions (10nM R1881 for 48

*Figure 3 continued on next page*

*Figure 3 continued*

hr). Biological triplicate samples were used for CWR22RV1, PNT2 and LNCaP cells, and technical replicate samples were used for RNAs prepared from PC3 cells. PSI levels were measured using RT-PCR analysis averaged from three replicates (mean data given in *Figure 3—source data 2*), and clustered in the heat map according to splicing patterns across the different conditions. The heatmap was generated using heatmap.2 function using R's 'gplots' package. The black shading in the heatmap denotes non-detection of the mRNA isoform after RT-PCR, and white denotes no change detected.

DOI: https://doi.org/10.7554/eLife.47678.009

The following source data is available for figure 3:

**Source data 1.** Alternative splicing events identified by Suppa2 (*Trincado et al., 2018*).
DOI: https://doi.org/10.7554/eLife.47678.010

**Source data 2.** Details of 44 experimentally validated ESRP1/ESRP2 target exons identified within prostate cancer cell lines.
DOI: https://doi.org/10.7554/eLife.47678.011

the splicing switches induced by ectopic expression of either ESRP2 or ESRP1 were reciprocal to the splicing switches detected after siRNA depletion of ESRP1/ESRP2 (*Figure 3*). Experimentally validated ESRP-regulated exons fell into two groups. Splicing of one group of exons were repressed by ectopic expression of ESRP1 or ESRP2 in PC3 cells, and reciprocally activated by endogenous ESRP1/ESRP2 depletion in LNCaP cells (these exons are in the top of the heatmap in *Figure 3*, from *OSBL3* to *FN1*). Splicing of the second group of exons were activated by ectopic expression of ESRP1 or ESRP2, and reciprocally repressed by ESRP1/ESRP2 depletion (from *TRIP10* to *ITGA6* in *Figure 3*).

## An androgen steroid hormone-ESRP2 axis controls alternative splicing in AR-positive prostate cancer cells

The above data identified a robust panel of alternative exons within prostate cancer cells that responded to ESRP1/ESRP2 expression levels. We next tested if this panel of ESRP2-regulated exons are additionally regulated by ambient androgen concentrations. LNCaP cells were harvested after growth in steroid deplete media and after 48 hr of androgen stimulation (this timing was designed to enable full levels of androgen-mediated ESRP2 protein induction, *Figure 1H*). Our prediction was that androgen stimulation of LNCaP cells would activate ESRP2 expression to regulate our panel of endogenous test exons. If this was the case, splicing switches in response to androgen stimulation should occur in a reciprocal direction to splicing changes induced by ESRP1/ESRP2 protein depletion in LNCaP cells. Consistent with these expectations, more than 70% (37/44) exons in our test panel demonstrated androgen regulated splicing (*Figure 3—source data 2*). Importantly, plotting the percent spliced-in (PSI) for each exon after 48 hr androgen stimulation (Y axis) versus the PSI after ESRP1/ESRP2 depletion (X axis) showed a significant negative correlation (slope = $-0.66$, $R^2 = 0.64$, p<0.0001) (*Figure 4A*). Thus, exons that showed more exon skipping in response to ESRP1/ESRP2 depletion had higher splicing inclusion after androgen stimulation (which would induce ESRP2 expression) (examples shown in *Figure 4A and B*). Reciprocally, exons that showed higher splicing inclusion in response to ESRP1/2 depletion also had less splicing inclusion after androgen stimulation (examples shown in *Figure 4A and C*). These results experimentally support an androgen-ESRP2 axis that controls splicing patterns in prostate cancer cells.

The genes containing ESRP-activated exons that were also activated by androgen exposure (*Figure 4B*) included: *MINK1* (exon 18) which encodes a pro-migratory serine/threonine kinase; *MAP3K7* (exon 12) which encodes a serine/threonine kinase that regulates signalling and apoptosis, activates NFKappaB, and is lost in aggressive prostate cancer (*Kluth et al., 2013*; *Rodrigues et al., 2015*); *GRLH1* (exon 5) that encodes a transcription factor involved in epithelial cell functions (*Jacobs et al., 2018*); and *FLNB* (exon 30), alternative splicing of which has been identified as a key switch contributing to breast cancer metastasis (*Li et al., 2018*; *Ravipaty et al., 2017*). Amongst the genes containing ESRP2-repressed exons that were also skipped in response to androgen stimulation (*Figure 4C*) were *DOCK7* (exon 23), which encodes a guanine nucleotide exchange factor involved in cell migration (*Gadea and Blangy, 2014*); and *RPS24* (exon 5), a gene that is highly expressed in prostate cancer (*Arthurs et al., 2017*).

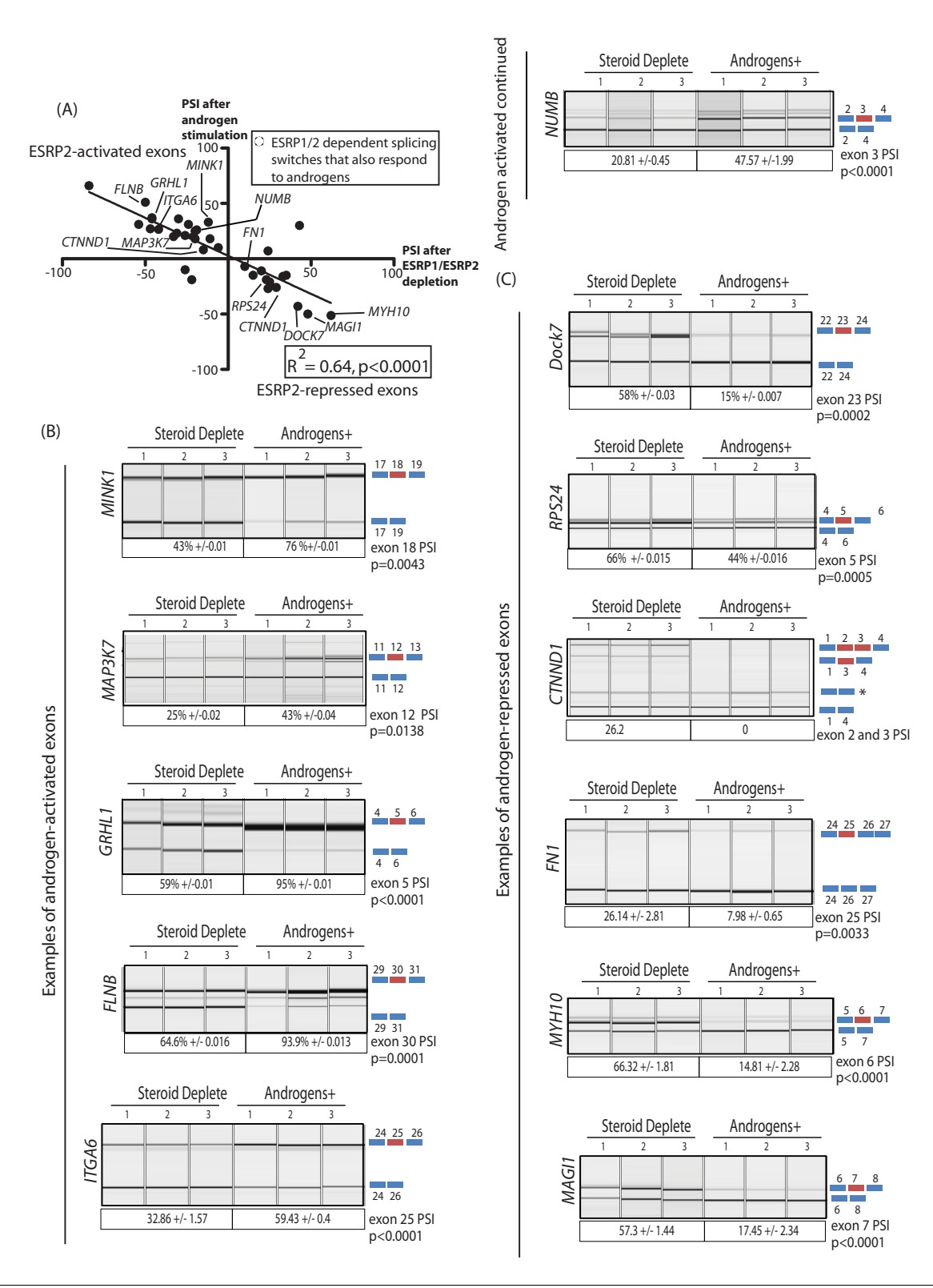

**Figure 4.** An androgen steroid hormone-ESRP2 axis controls alternative splicing in prostate cancer cells. (A) ESRP2-regulated exons are frequently also controlled by androgens in prostate cancer cells. 31/48 of the ESRP target exons (identified by RNAseq analysis of LNCaP cells depleted of ESRP1 and ESRP2) were regulated in the opposite direction in LNCaP cells treated by androgens (10nM R1881) for 48 hr (which would induce ESRP2 protein expression). Plotting the splicing responses to androgen stimulation with those after ESRP1/ESRP2 depletion revealed a negative correlation

*Figure 4 continued on next page*

Figure 4 continued

(slope = −0.66+/−0.09, Rsquare = 0.64, p<0.0001, calculated using Graphpad). Individual values for this graph are given in *Figure 3—source data 2*, and are averages from three biological replicates. (B) Capillary gel electrophoretograms showing splicing patterns of 3 biological replicates grown in steroid deplete media, or steroid deplete media supplemented with R1881, for alternative exons that are activated by androgen exposure in the *MINK1, MAP3K1, GRHL1, FLNB, ITGA6* and *NUMB* genes. (C) Capillary gel electrophoretograms showing splicing patterns of 3 biological replicates grown in steroid deplete media, or steroid deplete media supplemented with R1881, for alternative exons that are repressed by 48 hr androgen exposure in the *DOCK7, RPS24, CTNND1, FN1, MYH10* and *MAGI1* genes that were repressed by four androgen treatment. For parts (B–C) the p values were calculated using unpaired t tests, apart for *CTNND1* where zero inclusion of exons 2 and 3 were detected in the presence of androgens. For *CTNND1*, an RT-PCR product derived from a splice variant joining exons 1–4 via an alternative splice site is asterisked.
DOI: https://doi.org/10.7554/eLife.47678.012

## The AR-ESRP2 axis controls splicing of mRNA isoforms that are important for prostate cancer disease progression

To visualise the amplitude of ESRP2-mediated splicing control, we plotted PSIs measured in vitro after ectopic expression of ESRP1/ESRP2 versus PSI values after siRNA mediated depletion of ESRP1/ESRP2 (*Figure 5A*, using data from *Figure 3* and *Figure 3—source data 2*, slope = −0.74, $R^2$ = 0.6221, p<0.0001). Consistent with the heat map (*Figure 3*), ESRP2-regulated exons fell into two groups. Splicing of one group of exons were ESRP2-activated, and splicing of these were conversely repressed by ESRP1/ESRP2 depletion, while the second group of ESRP2-repressed exons had the reverse properties.

To assess how important ESRP2-regulated mRNAs might be in prostate cancer we monitored associated data for time taken to first biochemical tumour recurrence available in the TCGA PRAD cohort, in which information for 38/44 ESRP-regulated exons was available. This analysis revealed 3 groups of ESRP-regulated exons with different clinical associations. The group of ESRP1/ESRP2-promoted splice isoforms that correlated with decreased time to biochemical recurrence are shown in black on *Figure 5A* (individual plots are shown in *Figure 5—figure supplement 1*, and the functions of these genes and their associated splice isoforms in *Figure 5—source data 1*). Skipping of *RPS24* exon five correlates with a worse prognosis, and is the splice isoform promoted by ESRP2. Splicing inclusion of *RPS24* exon five is needed to maintain the *RPS24* open reading frame (*Wang et al., 2015*). Splicing inclusion of *NUMB* exon three also correlated with a worse prognosis, and is activated by ESRP2. *NUMB* exon three encodes peptide information enabling protein interactions between NUMB and MDM2, a protein that influence p53 stability (*Colaluca et al., 2018*).

Expression of the second group of ESRP1/ESRP2-promoted mRNA isoforms correlated with an increased time to biochemical occurrence. These exons are shown in green in *Figure 5A*, and include exons in the *FLNB, SLK* and *ITGA6* genes (functions of these genes and exons are summarised in *Figure 5—source data 2*). For example, inclusion of *ITGA6* exon 25 is activated by ESRP2, and predicted to alter signalling pathways activated by the encoded protein (*Groulx et al., 2014*). Splicing inclusion of the third set of exons did not correlate with time to biochemical recurrence (identified as grey dots in *Figure 5A*, and summarised in *Figure 5—source data 3*). These exons included *GRHL1* exon 5, splicing of which is needed to maintain the *GRHL1* reading frame. *GRHL1* encodes a transcription factor important for the operation of epithelial enhancer sequences (*Cieply et al., 2016*; *Jacobs et al., 2018*).

To provide some measurement of the enrichment for clinically-relevant events, we compared the significance of optimal biochemical reoccurrence (BCR) survival difference between ESRP-regulated and all other exons whose PSI variance across TCGA primary tumours was ≥0.005 (approximately the minimum for regulated events, to avoid biasing the potential relevance towards these). As illustrated in the violin plot in *Figure 5D*, there was a significant trend for a stronger prognostic value amongst the ESRP-regulated exons.

Further analysis of the PRAD cohort revealed that 19/38 ESRP-regulated exons also have different patterns of splicing inclusion between tumour and normal tissue (*Figure 5E* and *Figure 3—source data 2*). These differentially spliced exons include the AR-ESRP2-controlled alternative exons in the *DOCK7* and *RPS24* genes (both of which were excluded in prostate tumours compared to normal prostate tissue); and the alternative exons in the *MINK1* and *MAP3K7* genes (each of which had increased levels of splicing inclusion in prostate tumours compared to normal tissue). Further qRT-

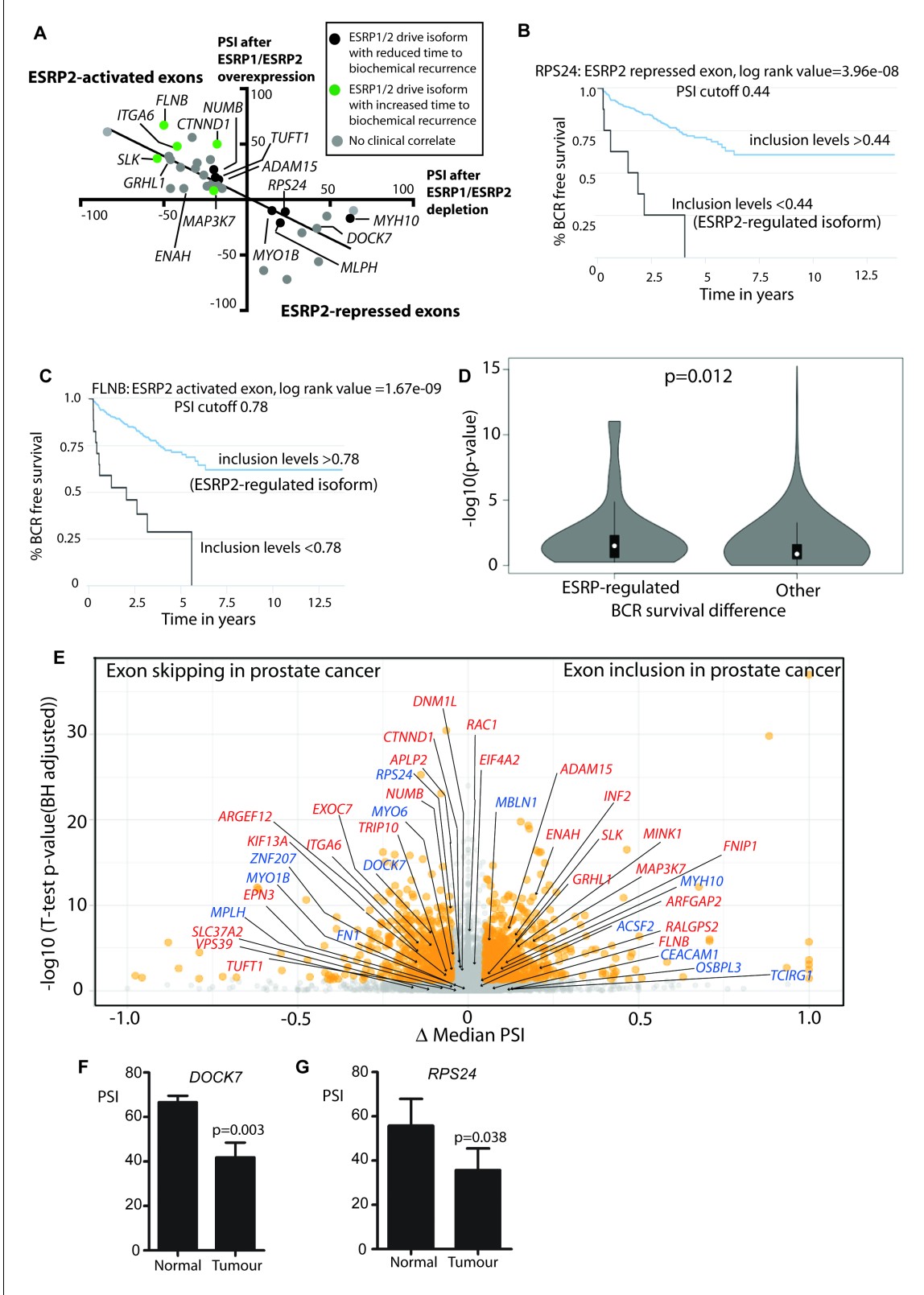

**Figure 5.** Alternative splicing patterns controlled by the androgen steroid hormone-ESRP2 splicing axis are clinically relevant for disease progression. (**A**) Graphical representation of levels of average PSI levels in response to ectopic ESRP2 expression in PC3 cells (Y axis) versus after ESRP1/ESRP2 depletion in LNCaP cells. Individual PSI values to make this graph are averaged from three biological replicates, and are given in *Figure 3—source data 2*. Note that the PSI values for ESRP over-expression refer to ESRP2 over-expression, with the exception of *FNIP1* and *SLC37A2* that are for ESRP1

*Figure 5 continued on next page*

*Figure 5 continued*

over-expression (see *Figure 3—source data 2*). Linear regression analysis of this data was analysed using Graphpad. Individual splice forms were correlated with clinical data for time to PSA biochemical recurrence within the PRAD cohort (*Saraiva-Agostinho and Barbosa-Morais, 2019*). Points on this graph corresponding to individual ESRP-regulated splice isoforms are coloured differently according to whether they correlate with an increased time to PSA biochemical recurrence (blue dots), a decreased time to biochemical recurrence (red dots) or had no significant correlation (black dots) is shown. (**B–C**) Kaplan-Meier plots showing data from TCGA PRAD cohort of percentage of tumours that are free of biochemical recurrence versus time in years, associated with expressing the alternative splice isoforms of (**B**) *RPS24* exon 5 (PSI cut off 0.44), and (**C**) *FLNB* exon 30 (PSI cut off 0.78) (*Saraiva-Agostinho and Barbosa-Morais, 2019*). (**D**) Violin plots of distributions of significance of biochemical recurrence (BCR) free survival difference between better and worse prognosis groups of patients defined by PSI cut-off values in primary tumour samples, for ESRP-regulated and all other exons whose PSI variance across TCGA primary tumours was ≥0.005 (approximately the minimum for regulated events, to avoid biasing the potential functional relevance towards these). The survival difference significance for each exon is given by -log10 of the p-value of the log-rank test used to compare survival distributions and the PSI cut-off value is the one minimising that p-value, given by *psichomics* (*Saraiva-Agostinho and Barbosa-Morais, 2019*).The depicted statistical significance (**p**) of the difference between the distributions summarised by the violins was calculated with a two-sided Wilcoxon signed-rank test. (**E**) Volcano plot showing alternative splicing analysis (*Saraiva-Agostinho and Barbosa-Morais, 2019*) of RNAseq data performed between normal prostate tissue and prostate tumour tissue from the TCGA PRAD cohort (consisting of 497 prostate tumour samples and 52 normal tissue). The t-test p-value (Benjamini-Hochberg adjusted for multiple testing) was used as metric of statistical significance. Significantly differentially spliced events (|Δ median PSI| ≥ 0.1 and FDR ≤ 0.01) are highlighted in orange, others in grey. ESRP2-activated exons are labelled in red text, and ESRP2-repressed exons are labelled in blue text. (**F**) Percentage splicing inclusion (PSI), quantified by RT-PCR, of *DOCK7* exon 23 within samples of prostate tumour and adjacent normal tissue (statistical significance calculated using t test). (**G**) Percentage splicing inclusion (PSI), quantified by RT-PCR, of *RPS24* exon 2 within nine matched samples of prostate tumour and adjacent normal tissue (statistical significance calculated using t test).
DOI: https://doi.org/10.7554/eLife.47678.013

The following source data and figure supplements are available for figure 5:

**Source data 1.** Properties of ESRP-regulated exons that correlate with a decreased time to biochemical recurrence.
DOI: https://doi.org/10.7554/eLife.47678.016

**Source data 2.** Properties of ESRP-regulated exons that correlate with an increased time to biochemical recurrence.
DOI: https://doi.org/10.7554/eLife.47678.017

**Source data 3.** Properties of ESRP-regulated exons that show no significant correlation with time to biochemical recurrence.
DOI: https://doi.org/10.7554/eLife.47678.018

**Figure supplement 1.** Kaplan-Meier plot showing data from TCGA PRAD cohort of percentage of tumours that are free of biochemical recurrence versus time in years associated with expressing ESRP2-regulated alternative splice isoforms.
DOI: https://doi.org/10.7554/eLife.47678.014

**Figure supplement 2.** Violin plot showing PSI levels for (**A**) *NUMB* exon 6, (**B**) *ITGA6* exon 25 (**C**) *RAC1* exon 3A, (**D**) *RPS24* exon 6, and (**E**) *MYO1B* exon 23, and (**F**) *FLNB* exon 31 in different grade prostate tumours.
DOI: https://doi.org/10.7554/eLife.47678.015

PCR analysis of an independent cohort confirmed more frequent skipping of *DOCK7* (exon 23) and *RPS24* (exon 5) in prostate tumour tissue compared to normal prostate (*Figure 5F and G*).

Some exons had more subtle changes than would be apparent from just comparing overall exon skipping and exon inclusion in prostate cancer. *NUMB* exon three and *ITGA6* exon 25 (both activated by ESRP2) are predominantly skipped in prostate tumours compared to normal tumour tissue, yet their PSI levels increase in larger, more advanced tumours to produce their respective mRNA isoforms that are associated with a decreased time to biochemical recurrence (*Figure 5—figure supplement 2A and B*). *RAC1* exon 3A (activated by ESRP2) falls into the 'grey' area when comparing inclusion in normal versus prostate cancer, but more detailed analysis show that this exon is highly included in higher Gleason grades of prostate cancer, again to produce the *RAC1* splicing isoform associated with a decreased time to biochemical recurrence (*Figure 5—figure supplement 2C*). *RPS24* exon 5 (repressed by ESRP2, and overall more skipped in tumours) is skipped more in larger more advanced tumours, making the mRNA isoform associated with a decreased time to biochemical recurrence (*Figure 5—figure supplement 2D*). Similarly, *MYO1B* exon 23 (skipped in response to ESRP2) is both overall more skipped in prostate tumour versus normal, and more skipped in higher Gleason grade cancers (*Figure 5—figure supplement 2E*). FLNB exon 31 (activated by ESRP2) actually shows slightly reduced splicing inclusion in larger, more aggressive tumours (*Figure 5—figure supplement 2F*).

## Splicing of key exons are switched by a drug that antagonises AR activity

The above data identified a subset of ESRP2-regulated splicing switches that associated with biochemical recurrence of prostate cancer after treatment. Since ESRP2 expression was repressed by ADT in patient prostate cancer tissue, we next investigated whether AR inactivation may influence mRNA splice isoforms that correlate with cancer progression. To test this, androgen induction of *ESRP2* mRNA expression was blocked using the androgen antagonist bicalutamide (Casodex) (*Figure 6A*). Consistent with Casodex preventing expression of some potentially harmful isoforms in prostate cancer cells, the splicing inclusion of *NUMB* exon three and *TUFT1* exon two were reduced by Casodex (both these exons are normally activated by androgen exposure and ESRP2). Likewise, exon skipping events in the *RPS24*, *FN1* and *MYH10* genes that correlated with a poorer prognosis were also reduced by Casodex (these exons are normally skipped in response to ESRP2). Not all the splicing switches induced by Casodex correlate with increased time to biochemical recurrence. Skipping of *CTNND1* exon 2 and 3 correlates with a decreased time to biochemical recurrence within the TGCA dataset (*Figure 5—figure supplement 1*), and this is the mRNA isoform promoted by Casodex treatment (*Figure 6C*). Splicing inclusion of *MAGI1* exon 7 (normally repressed by ESRP2) and *RALGPS2* exon 15 were also increased by Casodex treatment (*Figure 6B and C*).

ESRP2 and ESRP1 are important to maintain epithelial splicing programmes. We thus considered whether by repressing ESRP2 expression, ADT might also inadvertently switch splicing towards mesenchymal patterns that could facilitate metastasis. Consistent with this prediction, treatment of LNCaP cells with Casodex reduced splicing inclusion levels of the *FLNB* gene exon 30 by almost 20% (*Figure 6B*). Although it is not differentially spliced between normal prostate and prostate cancer (*Figure 5E*), increased skipping of *FLNB* exon 30 has been recently reported as a key driver of EMT in breast cancer development (*Li et al., 2018*). Similarly, Casodex treatment also increased splicing inclusion of what are normally mesenchymal-expressed exons in the *CTNND1* gene (*Warzecha et al., 2009b*) (*Figure 6C*).

We used siRNA as a further strategy to reduce AR expression (*Figure 6—figure supplement 1A*). As predicted, ESRP2 protein expression was reduced by siRNA depletion of the AR (*Figure 6—figure supplement 1A*). Furthermore, siRNA-mediated depletion of AR reduced levels of *FLNB* splicing inclusion from 84% to 69%, and levels of *TUFT1* exon 2 splicing from 23% to 9% (*Figure 6—figure supplement 1B*). Both these data support a scenario where splicing inclusion of ESRP2-dependent exons are controlled by expression levels of the AR.

## Splicing patterns respond to changes in the expression of ESRP2 alone

The above data suggested a model where decreases in ESRP2 expression in response to inhibition of AR activity are sufficient to induce splicing changes, even though ESRP1 was still expressed. To further investigate whether loss of ESRP2 alone would be sufficient to induce splicing changes we carried out individual siRNA-mediated depletion of ESRP2 both within both LNCaP and CWR22RV1 cells. Consistent with our model, single ESRP2 depletion was able to switch splicing patterns of exons within the *MAP3K7*, *ARFGAP2* and *CTNND1* genes (*Figure 6—figure supplement 2*; *Figure 3—source data 2*). As examples, individual depletion of ESRP2 reduced splicing inclusion of *MAP3K7* exon 12, and activated splicing inclusion of *CTNND1* exons 2 and 3 (*Figure 6 – Figure 3—source data 2*). Furthermore, splicing patterns of ESRP1/ESRP2 target exons were also responsive to single up-regulation of either ESRP1 or ESRP2 (*Figure 3—source data 2*).

## Discussion

In this study we report a novel molecular mechanism that explains how androgen steroid hormones control splicing patterns in prostate cancer cells, and unifies the functions of the AR both as a transcription factor and being able to control splicing. In this model, the AR controls expression of the master splicing regulator protein ESRP2, which then regulates the splicing patterns of key genes important for prostate cancer biology (*Figure 7*). Amongst the key data supporting this proposed model, we find that *ESRP2* is a direct and early target for transcriptional activation by the AR in prostate cancer cells. Furthermore, endogenous splice isoform patterns controlled by ESRP1 and ESRP2 also respond to androgen stimulation, siRNA-mediated depletion of the AR and/or the AR inhibitor

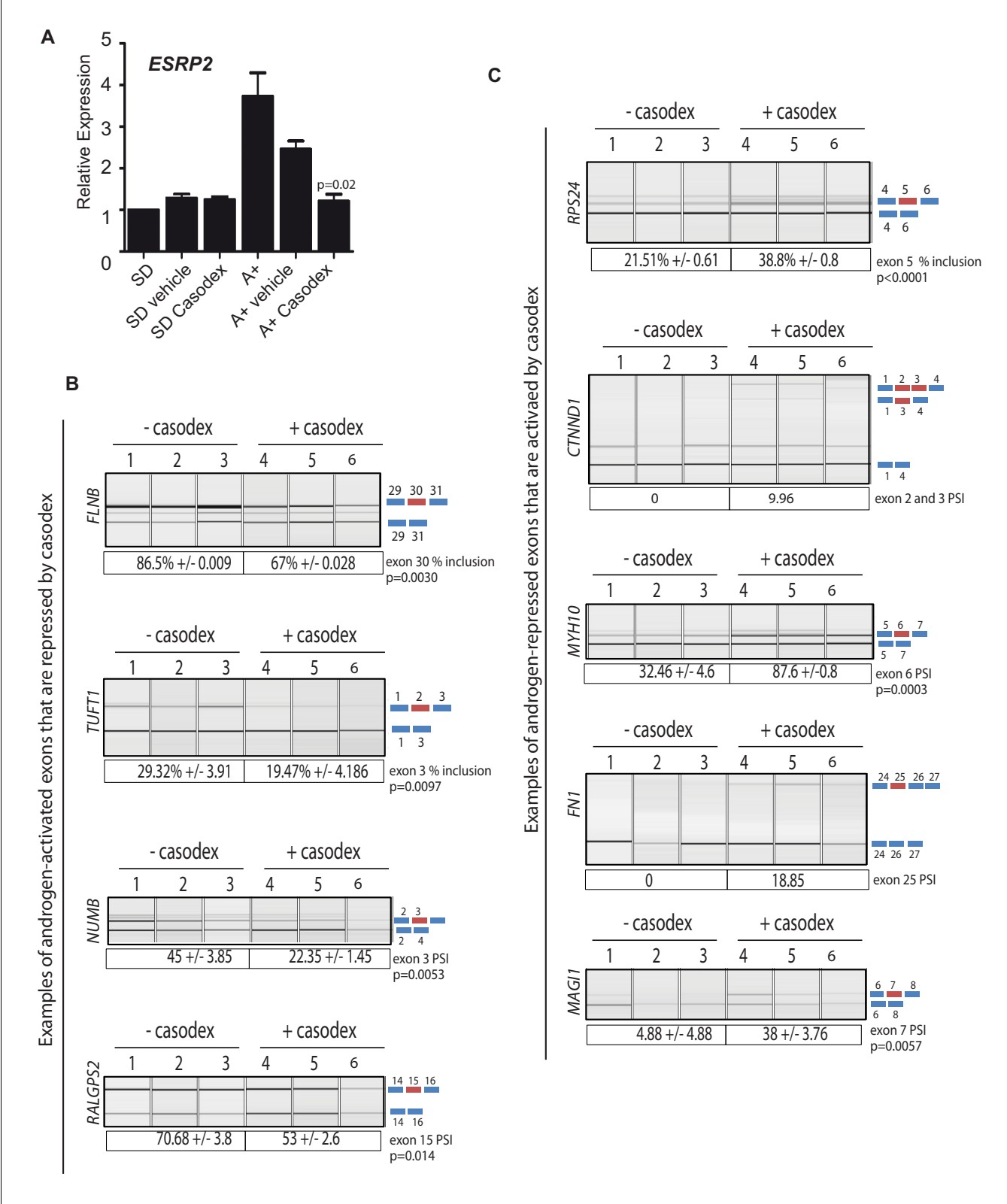

**Figure 6.** Pharmacological inhibition of AR function switches ESRP2-dependent splicing patterns. (**A**) *ESRP2* mRNA expression in cells grown in steroid deplete (SD) conditions, and after addition of androgens (A+) (quantified by real-time PCR from three biological replicates). Androgen-mediated activation of *ESRP2* expression was inhibited in the presence of 10 µM of the anti-androgen bicalutamide (Casodex). Cells were cultured for 24 hr. The p value shows the statistical significance that was calculated using a t test between the A + vehicle and the A + casodex samples. (**B–C**) Capillary gel

*Figure 6 continued on next page*

*Figure 6 continued*

electrophoretogram showing RT-PCR analysis the splicing response + /- 24 hr Casodex treatment for exons that are normally (**B**) activated or (**C**) repressed by androgens (three biological samples shown, statistical significances were calculated using a t test, with the exception of *CTNND1* where there was zero inclusion detected for exons 2 and 3 before depletion of ESRP2, and FN1 where there was zero inclusion of exon 25 before Casodex treatment).

DOI: https://doi.org/10.7554/eLife.47678.019

The following figure supplements are available for figure 6:

**Figure supplement 1.** siRNA depletion of the AR switches splicing of ESRP2- regulated exons.

DOI: https://doi.org/10.7554/eLife.47678.020

**Figure supplement 2.** siRNA depletion of ESRP2 alone is sufficient to switch splicing patterns of ESRP-regulated exons.

DOI: https://doi.org/10.7554/eLife.47678.021

bicalutamide (Casodex). While intuitively straightforward, this model is conceptually different from the mechanisms through which estrogen and progesterone have been shown to regulate splicing (via recruitment of splicing regulators as transcriptional cofactors, and by modulation of transcription speeds and chromatin structure).

Androgens are already known to substantially modify transcriptional levels in prostate cancer, with important implications for cell behaviour and cancer progression (*Munkley et al., 2016*). The data presented here imply that androgens also have an important role in controlling splicing patterns, particularly those that relate to epithelial/mesenchymal functions. Previous studies identified just a small number of alternative exons that are controlled by androgens in prostate cancer cells, none of which overlapped with the current study (*Munkley et al., 2018*; *Rajan et al., 2011*). We suggest that an important reason for this discrepancy is because previously splicing patterns were monitored after 24 hr of androgen exposure. Since we now show that splicing regulation by androgens operates indirectly through transcriptional control of ESRP2, 24 hr androgen exposure would not be sufficient to upregulate ESRP2 levels. In the current study we analysed androgen-dependent splicing switches after 48 hr, to allow sufficient time for ESRP2 induction at the protein level and re-equilibration of downstream splice isoform ratios. ESRP1 expression levels also decreased in prostate tumours following ADT so might also be under androgen-control in tissue, although did not reciprocally increase following androgen stimulation of cultured cells.

ESRP1 has recently been shown to be amplified in an aggressive subgroup of early onset prostate cancer, but how this contributes to disease progression has been not well understood (*Gerhauser et al., 2018*). Our data here show that ESRP1 and ESRP2 control a number of individual mRNA splice isoforms that correlate with time to biochemical recurrence (*Figure 5—figure supplement 2* and *Figure 5—figure supplement 1*), including of *MAP3K7* exon 12 inclusion which is associated with a shorter time to biochemical reoccurrence in the TGCA database. Deletion of the *MAP3K7* gene occurs in 30–40% of prostate tumours, and is associated with a poor clinical prognosis (*Goodall et al., 2016*; *Kluth et al., 2013*; *Liu et al., 2007*; *Wu et al., 2012*). *MAP3K7* is a key gene in prostate cancer, and *MAP3K7* exon 12 splicing is associated with epithelial properties of prostate cancer cells (*Dittmar et al., 2012*). More generally, epithelial splicing patterns may play an important role early in prostate cancer development in establishing primary tumours (*Figure 7*).

The expression of ESRPs appears to be plastic during cancer progression (*Hayakawa et al., 2017*; *Ishii et al., 2014*; *Ueda et al., 2014*). ESRPs have previously been shown to have a dual role in carcinogenesis with both gain and loss associated with poor patient prognosis (*Hayakawa et al., 2017*). ESRP1 expression is linked to poor survival and metastasis in lung cancer (*Yae et al., 2012*), and both ESRP1 and ESRP2 are upregulated in oral squamous cell carcinoma relative to normal epithelium (*Ishii et al., 2014*). Since ESRP2 is a critical component of epithelial-specific splicing programmes, we suggest that down-regulation of ESRP2 levels in response to ADT could dampen epithelial splicing patterns, helping to prime prostate cancer cells for future mesenchymal development and possibly contribute to development of metastasis. Supporting this, mesenchymal splicing patterns were induced by bicalutamide (Casodex) treatment of LNCaP cells, including in the *FLNB*, *CTNND1* and *MAP3K7* genes. *FLNB* encodes an actin binding protein which is linked to cancer cell motility and invasion (*Del Valle-Pérez et al., 2010*; *Iguchi et al., 2015*). Skipping of *FLNB* exon 30 is sufficient to initiate metastatic progression in breast cancer (*Li et al., 2018*). In experiments reported here androgens promote the FLNB isoform that is not associated with metastasis. Expression of the

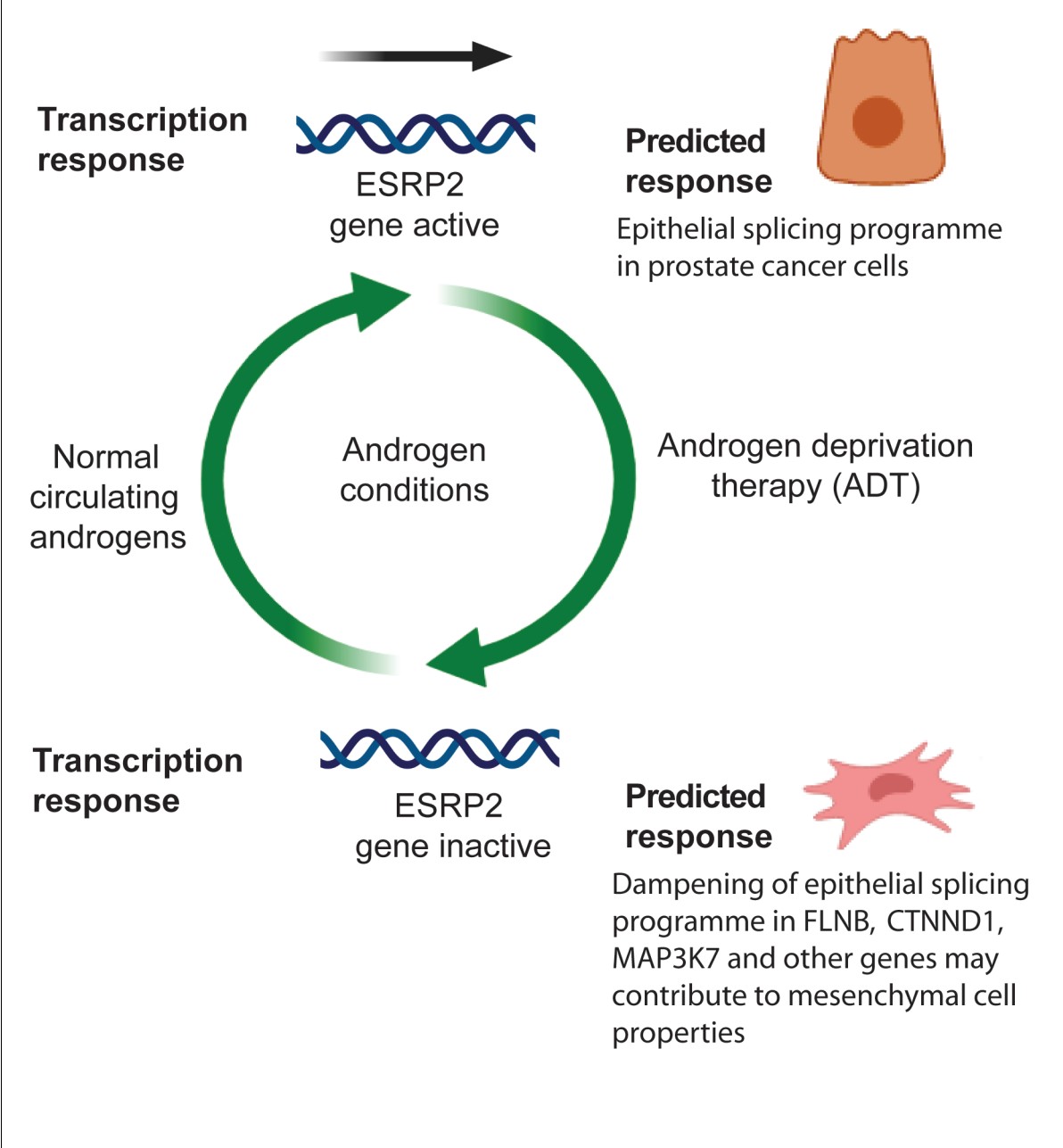

**Figure 7.** Model describing how exposure to androgens regulates splicing patterns in prostate cancer cells. Androgen exposure leads to transcription of the gene encoding the master splicing regulator protein *ESRP2*. This promotes epithelial splicing patterns within prostate cancer cells. Lower levels of circulating androgens after ADT lead to transcriptional repression of *ESRP2*. This results in a dampening of epithelial splicing patterns, and production of normally mesenchymal splice patterns including for the *FLNB*, *CTNND1* and *MAP3K7* genes. This image was created using BioRender.
DOI: https://doi.org/10.7554/eLife.47678.022

metastatic *FLNB* variant is promoted by bicalutamide (Casodex) treatment. In breast cancer, the metastatic effects of FLNB alternative splicing are mediated via the FOXC1 transcription factor. The role FOXC1 plays in prostate cancer progression is unknown, but FOXC1 expression may be linked to androgen receptor levels (*van der Heul-Nieuwenhuijsen et al., 2009*). ESRP2 also promotes skipping of epithelial-expressed exons in the *CTNND1* gene (catenin delta 2, encoding a protein involved in cell adhesion and signalling), while Casodex treatment induces expression of a normally mesenchyme-specific splice isoform (*Warzecha et al., 2009b*). The Map3k12Δexon12 splice isoform

is produced in response to ESRP2 depletion, and is usually expressed in highly metastatic cancer cell lines (*Tripathi et al., 2019*).

The clinical prognosis of metastatic prostate cancer is poor (*Livermore et al., 2016*). This makes the mechanisms that control metastasis of prostate cancer cells, and any links with ADT of prime importance. In prostate cancer EMT has been linked to a common mechanism underlying therapeutic resistance and is associated with poor prognosis (*Gravdal et al., 2007*). Sun et al. showed that although ADT can effectively control prostate tumour size initially, it simultaneously promotes EMT, an unintended consequence that could ultimately lead to CRPCa (*Sun et al., 2012*). Such direct links between ADT and EMT uncover an important yet overlooked consequence of the standard care treatment for advanced prostate cancer (*Byrne et al., 2016*). Although the causes of EMT in prostate cancer progression to CRPCa are likely to be complex, the down-regulation of ESRP proteins has been shown to be essential for EMT progression (*Horiguchi et al., 2012*). Thus, loss of ESRP expression may provide a molecular explanation why AR positive prostate cancer cells show increased susceptibility to EMT in response to ADT, and so is relevant to consider with regard to therapy. Our findings have important implications for second line treatment strategies in a clinical setting, and suggest an alternative approach may be to inhibit EMT in combination with ADT to prevent disease progression.

# Materials and methods

## Key resources table

| Reagent type (species) or resource | Designation | Source or reference | Identifiers | Additional information |
|---|---|---|---|---|
| Gene (*H. sapiens*) | *ESRP1* | | | |
| Gene (*H. sapiens*) | ESRP2 | | | |
| Cell line (*H. sapiens*) | LNCaP | ATCC | CRL-1740 | |
| Cell line (*H. sapiens*) | PC3 | ATCC | CRL-1435 | |
| Cell line (*H. sapiens*) | CWR-RV1 | ATCC | CRL-2505 | |
| Cell line (*H. sapiens*) | PNT2 | Sigma Aldrich | 95012613 | |
| Cell line (*H. sapiens*) | RWPE-1 | ATCC | CRL-11609 | |
| Antibody | Rabbit polyclonal anti-ESRP2 | Genetex | GTX123665 | 1:1000 dilution |
| Antibody | Rabbit polyclonal anti-ESRP1 | Sigma, | HPA023719 | 1:1000 dilution |
| Antibody | Mouse monoclonal anti-AR | BD Bioscience, | 554226 | 1:10000 dilution |
| Antibody | anti-actin rabbit polyclonal antibody | Sigma | A2668 | 1:2000 dilution |
| Antibody | anti-FLAG mouse monoclonal antibody | Sigma | F3165 | 1:2000 dilution |
| Antibody | normal rabbit IgG | Jackson labs | 711-035-152 | 1:2000 dilution |
| Antibody | normal mouse IgG | Jackson labs | 715-036-150 | 1:2000 dilution |

*Continued on next page*

*Continued*

| Reagent type (species) or resource | Designation | Source or reference | Identifiers | Additional information |
|---|---|---|---|---|
| Recombinant DNA reagent | ESRP1 plasmid | Gift from Prof Russ Carstens (University of Philadelphia. USA) | PIBX-C-FF-B-ESRP1 | |
| Recombinant DNA reagent | ESRP2 plasmid | Gift from Dr Keith Brown (University of Bristol. UK) | pBIGi hESRP2-FLAG | |
| Sequence based reagent | Primers to detect splice isoforms | This paper | | designed using Primer3 http://primer3.ut.ee/ |
| Sequence based reagent | qPCR primers | This paper | | designed using Primer3 http://primer3.ut.ee/ |
| Sequence based reagent | siRNAs | hs.Ri.ESRP1.13.1, hs.Ri.ESRP1.13.2, hs.Ri.ESRP2.13.1, hs.Ri.ESRP2.13.2, IDT (51-01-14-04), AR esiRNA EHU025951 Control esiRNA EHUEGFP Sigma | | |
| Commercial assay or kit | Rnaeasy plus kit | Qiagen | catalog number 74134 | |
| Commercial assay or kit | DNA free | Ambion | catalog number AM1906 | |
| Software, algorithm | Graphpad prism | https://graphpad.com | | |
| Chemical compound, drug | synthetic androgen analogue methyltrienolone (R1881) | Perkin–Elmer | NLP005005MG | 10 nM |
| Chemical compound, drug | Bicalutamide (Casodex) | Sigma | B9061 | 10 µM |

## Cell lines and cell culture

Cell culture and androgen treatment of cells was as described previously (*Munkley et al., 2015a*; *Munkley et al., 2015b*; *Munkley et al., 2015c*; *Munkley et al., 2014*; *Rajan et al., 2011*). All cells were grown at 37˚C in 5% CO2. LNCaP cells (CRL-1740, ATCC) were maintained in RPMI-1640 with L-Glutamine (PAA Laboratories, R15-802) supplemented with 10% Fetal Bovine Serum (FBS) (PAA Laboratories, A15-101). For androgen treatment of LNCaP cells, medium was supplemented with 10% dextran charcoal stripped FBS (PAA Laboratories, A15-119) to produce a steroid-deplete medium. Following culture for 72 hr, 10 nM synthetic androgen analogue methyltrienolone (R1881) (Perkin–Elmer, NLP005005MG) was added (Androgen +) or absent (Steroid deplete) for the times indicated. Similarly, LNCaP cells were pre-treated with with 10 µM bicalutamide (Casodex) or ethanol (vehicle) for 2 hr prior to addition of 10nM R1881 for 48 hr. Cell line validation was carried out using STR profiling was according to the ATCC guidelines. All cell lines underwent regular mycoplasma testing.

## Antibodies

The following antibodies were used for western blotting: Anti-ESRP2 rabbit antibody (Genetex, GTX123665), anti-rabbit ESRP1 (Sigma, HPA023719), anti-AR mouse antibody (BD Bioscience, 554226), anti-actin rabbit antibody (Sigma, A2668), anti-FLAG mouse monoclonal antibody (Sigma, F3165), normal rabbit IgG (711-035-152 Jackson labs) and normal mouse IgG (715-036-150 Jackson labs). For immunohistochemistry the following ESRP antibodies were tested: anti-rabbit ESRP1

(Sigma, HPA023719) and anti-rabbit ESRP2 (Abcam ab113486) but were found not to be specific for FFPE cell pellets.

## RT-qPCR

Cells were harvested and total RNA extracted using TRI-reagent (Invitrogen, 15596–026), according to the manufacturer's instructions. RNA was treated with DNase 1 (Ambion) and cDNA was generated by reverse transcription of 500 ng of total RNA using the Superscript VILO cDNA synthesis kit (Invitrogen, 11754–050). Quantitative PCR (qPCR) was performed in triplicate on cDNA using SYBR Green PCR Master Mix (Invitrogen, 4309155) using the QuantStudio 7 Flex Real-Time PCR System (Life Technologies). ESRP1 was detected using (ESRP1 for AGCACTACAGAGGCACAAACA; ESRP1 Rev TGGAGAGAAACTGGGCTACC). ESRP2 was detected using the primer combination (ESRP2 For CCT GAA CTA CAC AGC CTA CTA CCC; ESRP2 Rev TCC TGA CTG GGA CAA CAC TG). Samples were normalised using the average of three reference genes: GAPDH (GAPDH For AAC AGC GAC ACC CAT CCT C; GAPDH Rev TAGCACAGCCTGGATAGCAAC); β–tubulin (TUBB For C TTCGGCCAGATCTTCAGAC; TUBB Rev AGAGAGTGGGTCAGCTGGAA); and actin (ACTIN For CA TCGAGCACGGCATCGTCA; ACTIN Rev TAGCACAGCCTGGATAGCAAC).

## siRNA

siRNA mediated protein depletion of ESRP1/2 was carried out using Lipofectamine RNAiMAX Transfection Reagent (Thermo Fisher, 13778075) as per the manufacturer's instructions and for the times indicated. The siRNA sequences used were ESRP1 siRNA1 (hs.Ri.ESRP1.13.1); ESRP1 siRNA2 (hs.Ri.ESRP1.13.2); ESRP2 siRNA 1 (hs.Ri.ESRP2.13.1); ESRP2 siRNA 2 (hs.Ri.ESRP2.13.2); and a negative control siRNA (IDT (51-01-14-04)). AR esiRNA was as described previously (*Munkley et al., 2016*).

## Immunohistochemistry

Freshly cut tissue sections were analysed for immunoexpression using Ventana Discovery Ultra autostainer (Ventana Medical Systems, Tucson, Arizona). In brief, tissue sections were incubated in Cell conditioning solution 1 (CC1, Ventana) at 95℃ to retrieve antigenicity, followed by incubation with respective primary antibodies described above. Bound primary antibodies were visualized using UltraMap DAB anti-Rb Detection Kit.

## AR-ChIP

LNCaP cells were stimulated with 10 nM R1881 overnight. The ChIP assay was performed using the one step ChIP kit (Abcam ab117138) as per manufacturer's instruction. Briefly, cells were fixed and crosslinked in 1% formaldehyde for 10 min at 37℃ and incubated with protease inhibitors. Chromatin was isolated from cell lysates and enzymatically fragmented using an EZ-Zyme Chromatin Prep Kit (Merck 17 375). 10 ug of anti - AR antibody (Abcam ab74272) or IgG control antibody was used to precipitate DNA crosslinked with the androgen receptor. Enriched DNA was then probed by qPCR using primers targeting the ESRP2 regulatory region to assess AR binding intensity. Primer sequences used to detect PSA were (PSA ChIP for GCC TGG ATC TGA GAG AGA TAT CAT C; PSA Chip rev ACA CCT TTT TTT TTC TGG ATT GTT G). Primers used to detect AR binding near to ESRP2 were (ESRP2 Chip for TCCCGAGTAGCTGGGACTAC; ESRP2 Chip rev CAGTGGCTTACACC TGGGAG).

## Creation of PC3 stable cell lines

The ESRP1 plasmid (PIBX-C-FF-B-ESRP1) was a gift from Prof Russ Carstens (University of Philadelphia. USA) and the ESRP2 plasmid (pBIGi hESRP2-FLAG) from Dr Keith Brown (University of Bristol, UK). PC3 cells were transfected using FuGene HD Transfection Reagent as per manufacturer's instructions. Stable transfectants with ESRP1 was selected using 10 µg/ml Blasticidin and ESRP2 plasmid was selected using 150 ug/ml Hygromycin. ESRP2 Plasmid was inducible by 2.5 ug/ml doxycycline for 48 hr. PC3 ESRP1 overexpressed cells were transfected with pBIGi hESRP2-FLAG plasmid using the same protocol.

## In vitro cell proliferation analysis

For cell growth curves (carried out for in vitro analysis of PC3 stable cell lines), PC3 cells were seeded 100,000 cells per well in 12-well plate in eight plates. Cells were counted every 24 hr after seeding in the plate. All the treatments had 12 repeats. WST assays were carried out over 7 days as per manufacturer's instructions (Cayman, CAY10008883). For DU145 cells 10,000 cells were seeded per well in a 96 well plate. All data was tested by two-way ANOVA.

## RNAseq analysis

LNCaP cells (passage 19) were treated with either control siRNAs or siRNAs targeting ESRP1 and ESRP2 for 72 hr (samples prepared in triplicate). RNA was extracted 72 hr after siRNA treatment using the Qiagen RNAeasy kit (Cat No. 74104) as per the manufacturer's instructions. RNAseq was carried out using TruSeq Stranded mRNA Sequencing NextSeq High-Output to obtain 2 × 75 bp reads. Quality control of reads was performed using FastQC. Reads were mapped to the hg38 transcriptome using Salmon. Differential gene expression analysis was performed using DESeq2. Percent spliced-in (PSI) estimates for splicing events were calculated using SUPPA2 (*Trincado et al., 2018*) based on isoform transcripts per million (TPM) estimates from Salmon (*Patro et al., 2017*). Quantification utilised Gencode gene models (release 28). Differential PSI was calculated using DiffSplice using the empirical method (*Hu et al., 2013*). Events with a delta PSI > 10% and FDR < 0.05 were considered as significant.

## *Psichomics* and bioinformatic analysis of PRAD cohort

Clinical expression patterns of ESRP2-regulated exons were monitored using *psichomics* (*Saraiva-Agostinho and Barbosa-Morais, 2019*). Differential splicing analysis between primary solid tumour and solid tissue normal samples were subsequently performed to evaluate relative higher inclusion levels in either tumour or normal tissue samples using Δ median and t-test p-value (Benjamini-Hochberg adjusted) values. Survival analysis based on TCGA clinical data derived from prostate cancer patient samples was performed with time to first PSA biochemical recurrence being the event of interest. Additional statistical analyses and generation of plots were performed in *R* (*R Development Core Team, 2019*). Violin plots were created with R package *vioplot* (*Adler and Kelly, 2018*).

## Tumour xenografts

Stable overexpression of *ESRP1* and stable doxycycline-inducible overexpression of either *ESRP2* alone or *ESRP1* and *2* were obtained using PC3 cells (that have the low endogenous levels of both proteins). One million PC3 overexpressing *ESRP1* or control cells were injected subcutaneously in the flank of male nude mice and tumour volumes were monitored. Two million PC3 cells overexpressing *ESRP2*, overexpressing *ESRP1* and *2*, or control cells were injected subcutaneously in the flank of male nude mice and tumour volumes were monitored. PC3 ESRP2 and PC3 ESRP1/2 cells were cultured in medium supplemented with 2.5 ug/ml doxycycline for 48 hr prior to injecting into nude mice to induce ESRP2 expression and mice were administered Doxycycline repeatedly. Tumour diameters were measured using calipers.

## Clinical samples

Our study made use of RNA from 32 benign samples from patients with benign prostatic hyperplasia (BPH) and 17 malignant samples from transurethral resection of the prostate (TURP) samples. Malignant status and Gleason score were obtained for these patients by histological analysis. We also analysed normal and matched PCa tissue from nine patients obtained by radical prostectomy. The samples were obtained with ethical approval through the Exeter NIHR Clinical Research Facility tissue bank (Ref: STB20). Written informed consent for the use of surgically obtained tissue was provided by all patients. The RNA samples analysed in *Figure 2C* were previously published (*Walker et al., 2017*).

## Statistical analyses

All statistical analyses were performed using GraphPad Prism 6 (GraphPad Software, Inc). Statistical analyses were conducted using the GraphPad Prism software (version 5.04/d). PCR quantification of

mRNA isoforms was assessed using the unpaired student's t-test. Data are presented as the mean of three independent samples ± standard error of the mean (SEM). Statistical significance is denoted as *p<0.05, **p<0.01, ***p<0.001 and ****p<0.0001.

## Acknowledgements

This work was funded by Prostate Cancer UK [PG12-34, S13-020 and RIA16-ST2-011], Breast Cancer Now [2014NovPR355], and the BBSRC [BB/P006612/1]. The work performed at the Vancouver Prostate Centre was funded by the Terry Fox Research Institute (TFRI-NF-PPG). The authors would like to thank Dr Steven Walker, Dr Gemma Logan and Professor Richard Kennedy for kindly providing prostate cancer RNA samples from their 2017 *European Urology* (Walker, Knight, McCavigan, Logan, Berge, Sherif, Pandha, Warren, Davidson, Uprichard, Blayney, Price, Jellema, Steele, Svindland, McDade, Eden, Foster, Mills, Neal, Mason, Kay, Waugh, Harkin, Watson, Clarke, & Kennedy, 2017) paper for use in *Figure 2*. The ESRP1 plasmid was a gift from Prof Russ Carstens (University of Philadelphia, USA) and the ESRP2 plasmid from Dr Keith Brown (University of Bristol, UK). Collection of patient RNA samples was supported/funded by the NIHR Exeter Clinical Research Facility, but the opinions given in this paper do not necessarily represent those of the NIHR, the NHS or the Department of Health. The authors thank Dr Prabhakar Rajan (Barts Cancer Institute, London) for very helpful comments on the manuscript.

## Additional information

### Funding

| Funder | Grant reference number | Author |
|---|---|---|
| Prostate Cancer UK | PG12-34 | Jennifer Munkley<br>Emma Scott<br>Karen E Livermore |
| Biotechnology and Biological Sciences Research Council | BB/P006612/1 | Ingrid Ehrmann |
| Terry Fox Research Institute | TFRI-NF-PPG | Mads Daugaard |
| Breast Cancer Now | 2014NovPR355 | Caroline Dalgliesh |
| Prostate Cancer UK | RIA16-ST2-011 | Jennifer Munkley<br>Emma Scott<br>Karen E Livermore |
| BBSRC | BBSRC BB/J007293/2 | Sebastian Oltean |
| China Scholarship Council PhD fellowship | | Ling Li |

The funders had no role in study design, data collection and interpretation, or the decision to submit the work for publication.

### Author contributions

Jennifer Munkley, Conceptualization, Supervision, Investigation, Writing—original draft; Ling Li, Htoo Zarni Oo, Teresa Mendes Maia, Investigation, Writing—review and editing; S R Gokul Krishnan, Formal analysis, Investigation; Gerald Hysenaj, Emma Scott, Ingrid Ehrmann, Karen E Livermore, Investigation; Caroline Dalgliesh, Validation, Investigation, Writing—review and editing; Kathleen Cheung, Data curation, Investigation; Hanna Zielinska, Oliver Thompson, Bridget Knight, Paul McCullagh, John McGrath, Malcolm Crundwell, Lorna W Harries, Resources; Mads Daugaard, Nuno L Barbosa-Morais, Resources, Supervision, Investigation, Writing—review and editing; Simon Cockell, Resources, Data curation, Supervision; Sebastian Oltean, Conceptualization, Supervision, Investigation, Writing—review and editing; David J Elliott, Conceptualization, Supervision, Funding acquisition, Investigation, Writing—original draft, Project administration

## Author ORCIDs

Jennifer Munkley (iD) http://orcid.org/0000-0002-8631-4531
S R Gokul Krishnan (iD) http://orcid.org/0000-0003-4886-2710
Teresa Mendes Maia (iD) http://orcid.org/0000-0003-0038-9629
Nuno L Barbosa-Morais (iD) http://orcid.org/0000-0002-1215-0538
David J Elliott (iD) https://orcid.org/0000-0002-6930-0699

## Ethics

Human subjects: RNA samples from prostate cancer patients were obtained with ethical approval through the Exeter NIHR Clinical Research Facility tissue bank (Ref: STB20). Written informed consent for the use of surgically obtained tissue was provided by all patients.
Animal experimentation: Animal work was performed with the approval of Bristol University animal research ethics committee, according to recommendations of www.nc3rs.org.uk, and the UK Government Home Office (home office license PPL 30/3105). All experiments and procedures were approved by the UK Home office in accordance with the Animals (Scientific Procedures) Act 1986, and the Guide for the Care and Use of Laboratory Animals was followed.

## Decision letter and Author response

Decision letter https://doi.org/10.7554/eLife.47678.027
Author response https://doi.org/10.7554/eLife.47678.028

## Additional files

### Supplementary files

• Transparent reporting form
DOI: https://doi.org/10.7554/eLife.47678.023

### Data availability

Sequencing data have been deposited in GEO under accession code GSE129540.

The following dataset was generated:

| Author(s) | Year | Dataset title | Dataset URL | Database and Identifier |
|---|---|---|---|---|
| Munkley J, Elliott D, Cockell S, Cheung K | 2019 | RNAseq analysis of ESRP regulated splicing events in prostate cancer | http://www.ncbi.nlm.nih.gov/geo/query/acc.cgi?acc=GSE129540 | NCBI Gene Expression Omnibus, GSE129540 |

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
