## [Decision Letter]

Thank you for submitting your article "Androgen-regulated transcription of ESRP2 drives alternative splicing patterns in prostate cancer" for consideration by *eLife*. Your article has been reviewed by three peer reviewers, including Juan Valcárcel as the Reviewing Editor and Reviewer #1, and the evaluation has been overseen by James Manley as the Senior Editor.

The reviewers have discussed the reviews with one another and the Reviewing Editor has drafted this decision to help you prepare a revised submission.

Summary:

The manuscript from Munkley, Elliott and colleagues shows that the epithelial splicing regulator *ESRP2* is transcriptionally upregulated by the androgen receptor (AR), an observation based on a previous study of gene expression changes in response to androgen in the androgen receptor positive LNCaP prostate cancer cell line by some of these investigators. *ESRP2* upregulation leads to a series of changes in alternative splicing, including switches with potential effects in disease relapse and metastasis which correlate with disease outcomes. Prostate cancer is driven by androgens via AR, and therapy involves androgen deprivation (ADT) to slow progression. However, it has also been reported that ADT promotes epithelial mesenchymal transition (EMT) (e.g. Sun et al., 2012), which might be related to the common progression to castration resistant prostate cancer following ADT. Munkley et al. show that levels of ESRP2 are reduced after androgen deprivation in 7 prostate cancer patients. A number of other analyses using additional cell lines, a xenograft model, and data from other published prostate cancer samples leads to a general proposal that a decrease in *ESRP2* expression (but not *ESRP1*) and some splicing changes associated with its depletion following androgen deprivation may be associated with prostate cancer progression and worse outcomes. One highlighted example is exon 30 in *FLNB*, skipping of which is associated with metastatic progression in breast cancer.

A number of papers describing roles for *ESRP1/2* in various cancers including breast, colorectal, lung, and ovarian carcinomas have yielded conflicting conclusions on the role of ESRPs or epithelial-specific isoforms it regulates, such as CD44, in cancer progression and/or patient outcomes. In some cases ESRPs are proposed to be tumor suppressors, whereas in other cases they are proposed to promote more aggressive cancers (see, for example, Zhang et al., Genes and Dev 33: 166-179 and references therein). As cited by the authors, a recent manuscript reports that duplication and increased expression of *ESRP1* (which would largely promote the same splicing events as *ESRP2*) is associated with more aggressive human prostate cancers. Thus, a central question is whether the current manuscript can provide further clarity regarding the general role of ESRPs (including ESRP2) in cancer, including prostate cancer.

Munkley et al. raise the clinically-relevant point that current treatments for prostate cancer might have undesirable side-effects by inhibiting *ESRP2* mediated splicing events. Overall, the manuscript is clearly presented. The data documenting the ESRP and AR regulated splicing program, and the restriction of tumor growth by ESRPs (Figures 1-4, 6) are very clear with very nice correlations between responses to ESRP overexpression, knockdown and androgen stimulation.

Essential revisions:

1) A key concern relates to the relative levels and effects of ESRP1 and ESPR2 under conditions of androgen induction or ADT in prostate cells. The authors do a good job documenting that *ESRP2* is under transcriptional control of the androgen receptor, while *ESRP1* is not, and that there is a 2-fold reduction in *ESPR2* expression post-ADT in cancer samples. On the other hand, (a) both *ESRP1* and *ESRP2* seem down-regulated at the protein level in androgen receptor-negative prostate cancer cells lines (probably by different mechanisms), (b) both *ESRP1* and *ESRP2* mRNAs are up-regulated in tumor samples compared to controls, (c) both *ESRP1* and *ESRP2* are up-regulated in a cohort of metastatic patient samples, (d) the correlation between ESRP levels and recurrence free survival is a more significant for *ESRP1* than *ESRP2*, and (e) a number of functional assays from this manuscript and other publications argue that both *ESRP1* and *ESPR2* can contribute to regulate overlapping targets relevant for epithelial-specific splicing. Therefore one key question that remains is to what extent the androgen-mediated transcriptional regulation of *ESRP2* does contribute to splicing regulation in the context of the relative levels/activities of *ESRP1*: while a number of the results presented show that androgen treatment can promote splicing towards a stronger "epithelial" pattern, the authors should make additional efforts to demonstrate that ablation of *ESRP2* alone (in the presence of *ESRP1*) leads to substantial changes in splicing that would be expected to explain the association of a loss of *ESRP2* with worse outcomes, which is an essential point for the validity of their model. For example, an analysis similar to that of Figure 1A for *ESRP1* should be included, as well as other experiments aimed to determine whether the activity of *ESRP1* can buffer the effects of ATD on *ESRP2*.

2) There is also a need for clarity in terms of the coherence of the predicted biological effects of the alternative splice site switches and at least one proof-of-principle demonstration that they are relevant for any property of prostate cells relevant to cancer, as it is difficult to draw firm conclusions from the data presented as to whether the regulation of *ESRP2* by androgens is definitively associated with prostate cancer progression or outcomes in a positive or negative manner.

a) Figure 5A shows exons that are more included or skipped in prostate cancer vs. normal using TCGA data. But only 6 of the 44 ESRP-AR regulated events are highlighted on the plot, two of which do not change significantly, including FLNB which is highlighted in the Abstract and is the only event used to test the response to the AR antagonist Casodex. All of the events from Figure 3 should be highlighted in Figure 5A, with ESRP activated and repressed exons clearly distinguished by colour or symbol. The authors should explain – when known – the nature of the differential activities of the isoforms and whether the isoform switch observed in the presence of androgens/mediated by ESRPs is predicted to contribute, repress or be neutral to tumor cell growth, apoptosis, motility, metastasis, etc. and therefore whether a functionally coherent program of alternative splicing is coordinated by ERSPs or whether various contrasting contributions are predicted whose relative significance will depend on context, etc. If not, is it possible to stratify the data e.g. by tumor grade, or by ESRP expression level? Would this for instance, reveal different classes where events such as FLNB do show a difference between cancer and normal in some classes?

b) In Figure 6, why is *FLNB* e30 the only splicing event monitored for response to Casodex – especially since this is one of the events that is not altered between prostate cancer and normal tissue? This figure should be more systematic with more splicing events.

c) Increased inclusion of exon 30 in *FLNB* (which occurs for example upon androgen stimulation) is consistent with inhibition of EMT (something that could be stated more clearly in the text). But there is no mechanistic model presented as to how a change in *FLNB* splicing (or other targets) impacts prostate CA. What about the other alternative splicing events highlighted in Figures 4/5? Even if *FLNB* splicing switches have been shown to influence expression of EMT markers in breast cancer cells (Li et al., 2018), it will be essential to show that the degree of switch observed in prostate cancer cells (for *FLNB* or any other gene) has a relevant biological readout.

---

## [Author Response]

Essential revisions:1) A key concern relates to the relative levels and effects of ESRP1 and ESPR2 under conditions of androgen induction or ADT in prostate cells. The authors do a good job documenting that ESRP2 is under transcriptional control of the androgen receptor, while ESRP1 is not, and that there is a 2-fold reduction in ESPR2 expression post-ADT in cancer samples. On the other hand, (a) both ESRP1 and ESRP2 seem down-regulated at the protein level in androgen receptor-negative prostate cancer cells lines (probably by different mechanisms), (b) both ESRP1 and ESRP2 mRNAs are up-regulated in tumor samples compared to controls, (c) both ESRP1 and ESRP2 are up- regulated in a cohort of metastatic patient samples, (d) the correlation between ESRP levels and recurrence free survival is a more significant for ESRP1 than ESRP2, and (e) a number of functional assays from this manuscript and other publications argue that both ESRP1 and ESPR2 can contribute to regulate overlapping targets relevant for epithelial-specific splicing. Therefore one key question that remains is to what extent the androgen-mediated transcriptional regulation of ESRP2 does contribute to splicing regulation in the context of the relative levels/activities of ESRP1: while a number of the results presented show that androgen treatment can promote splicing towards a stronger "epithelial" pattern, the authors should make additional efforts to demonstrate that ablation of ESRP2 alone (in the presence of ESRP1) leads to substantial changes in splicing that would be expected to explain the association of a loss of ESRP2 with worse outcomes, which is an essential point for the validity of their model. For example, an analysis similar to that of Figure 1A for ESRP1 should be included, as well as other experiments aimed to determine whether the activity of ESRP1 can buffer the effects of ATD on ESRP2.

This is a good point. We had not previously considered androgen regulation of *ESRP1*, since only *ESRP2* originally was identified as a reciprocally regulated gene after acute androgen stimulation versus ADT. We now show that *ESRP1* mRNA levels do decrease in response to ADT (new Figure 1A), and confirm that only *ESRP2* mRNA levels increase in response to androgen stimulation of tissue culture cells. We also have carried out single siRNA depletion of *ESRP2*. This is sufficient to change splice isoform ratios (even in the presence of ESRP1) for 3 candidate events that we have tested (new Figure 6 —figure supplement 2). We also present data in Figure 3, where we have over-expressed either ESRP1 or ESRP2 alone in PC3 cells (which have low endogenous ESRP expression levels), and show that either can induce splicing responses.

2) There is also a need for clarity in terms of the coherence of the predicted biological effects of the alternative splice site switches and at least one proof-of-principle demonstration that they are relevant for any property of prostate cells relevant to cancer, as it is difficult to draw firm conclusions from the data presented as to whether the regulation of ESRP2 by androgens is definitively associated with prostate cancer progression or outcomes in a positive or negative manner.a) Figure 5A shows exons that are more included or skipped in prostate cancer vs. normal using TCGA data. But only 6 of the 44 ESRP-AR regulated events are highlighted on the plot, two of which do not change significantly, including FLNB which is highlighted in the Abstract and is the only event used to test the response to the AR antagonist Casodex. All of the events from Figure 3 should be highlighted in Figure 5A, with ESRP activated and repressed exons clearly distinguished by colour or symbol.

We now add all this information in the format requested (this updated volcano plot is now Figure 5E).

The authors should explain – when known – the nature of the differential activities of the isoforms and whether the isoform switch observed in the presence of androgens/mediated by ESRPs is predicted to contribute, repress or be neutral to tumor cell growth, apoptosis, motility, metastasis, etc. and therefore whether a functionally coherent program of alternative splicing is coordinated by ERSPs or whether various contrasting contributions are predicted whose relative significance will depend on context, etc.

We now add three supplementary tables that describe the known functions of ESRP2-regulated exons. These are the new Figure 5—source data 1 (ESRP2-regulated splices that correlate with a shorter time to biochemical recurrence); the new Figure 5—source data 2 (ESRP2-regulated splices that correlate with an increased time to biochemical recurrence); and the new Figure 5—source data 3(ESRP2-regulated splices that do not significantly correlate with time to biochemical recurrence). We discuss the possible functions of these exons on prostate tumours in the text and figure supplements (e.g. *RAC1* exon 3b is known from the literature to promote tumour cell migration).

If not, is it possible to stratify the data e.g. by tumor grade, or by ESRP expression level? Would this for instance, reveal different classes where events such as FLNB do show a difference between cancer and normal in some classes?

We include data now where we have stratified data by tumour grade (this is the new Figure 5—figure supplement 2). This shows that some exons (e.g. *FLNB, NUMB, ITGA6, RAC1* do show differences when tumour grades are considered)

b) In Figure 6, why is FLNB e30 the only splicing event monitored for response to Casodex – especially since this is one of the events that is not altered between prostate cancer and normal tissue? This figure should be more systematic with more splicing events.

We now make an updated Figure 6more systematic, containing electrophoretograms for more splicing events. We also include more extensive casodex data within an updated Figure 3—source data 2. Since the updated Figure 6 is now more extensive, we have moved the AR siRNA splicing figure to a new Figure 6—figure supplement 1. This AR siRNA splicing figure also contains many more examples that those we initially showed in our original manuscript.

c) Increased inclusion of exon 30 in FLNB (which occurs for example upon androgen stimulation) is consistent with inhibition of EMT (something that could be stated more clearly in the text). But there is no mechanistic model presented as to how a change in FLNB splicing (or other targets) impacts prostate CA. What about the other alternative splicing events highlighted in Figures 4/5? Even if FLNB splicing switches have been shown to influence expression of EMT markers in breast cancer cells (Li et al., 2018), it will be essential to show that the degree of switch observed in prostate cancer cells (for FLNB or any other gene) has a relevant biological readout.

We now discuss more clearly that increased inclusion of *FLNB* e30 is consistent with inhibition of EMT, and how in response to Casodex prostate cancer cells express mesenchymal isoforms of *FLNB, CTNND1* and *MAP3K7*. To address the function of these splicing switches we include three new supplementary tablesthat discuss the known biological readouts of androgen/ESRP-regulated exons (new Figure 5—source data 1-3). To address the potential impact of these splicing events in prostate cancer, we include Kaplan-Meier plots that correlate splicing switches with time to tumour biochemical recurrence (updated Figure 5 —figure supplement 1). We also only include now exons in these plots for which there are sufficient patients to draw firm conclusions about isoform levels relative to biochemical recurrence).